# Simple yet Effective Incomplete Multi-view Clustering: Similarity-level Imputation and Intra-view Hybrid-group Prototype Construction

**Shengju Yu** [1], **Zhibin Dong** [1], **Siwei Wang** [2]*, **Pei Zhang** [1], **Yi Zhang** [1], **Xinwang Liu** [1]*,
**Naiyang Guan** [3], **Tiejun Li** [1]*, **Yiu-ming Cheung** [4]
[1]National University of Defense Technology, [2]Intelligent Game and Decision Lab
[3]National Innovation Institute of Defense Technology, [4]Hong Kong Baptist University
`yu-shengju@foxmail.com {wangsiwei13,xinwangliu,tjli}@nudt.edu.cn`

## Abstract

Most of incomplete multi-view clustering (IMVC) methods typically choose to ignore the missing samples and only utilize observed unpaired samples to construct bipartite similarity. Moreover, they employ a single quantity of prototypes to extract the information of **all** views. To eliminate these drawbacks, we present a simple yet effective IMVC approach, SIIHPC, in this work. It firstly transforms partial bipartition learning into original sample form by virtue of reconstruction concept to split out of observed similarity, and then loosens traditional non-negative constraints via regularizing samples to more freely characterize the similarity. Subsequently, it learns to recover the incomplete parts by utilizing the connection built between the similarity exclusive on respective view and the consensus graph shared for all views. On this foundation, it further introduces a group of hybrid prototype quantities for each individual view to flexibly extract the data features belonging to each view itself. Accordingly, the resulting graphs are with various scales and describe the overall similarity more comprehensively. It is worth mentioning that these all are optimized in one unified learning framework, which makes it possible for them to reciprocally promote. Then, to effectively solve the formulated optimization problem, we design an ingenious auxiliary function that is with theoretically proven monotonic-increasing properties. Finally, the clustering results are obtained by implementing spectral grouping action on the eigenvectors of stacked multi-scale consensus similarity. Experimental results confirm the effectiveness of SIIHPC.

## 1 Introduction

Incomplete multi-view clustering (IMVC), a representative unsupervised learning approach, is grasping increasing concerns owing to its effectiveness in grouping heterogeneous data containing missing samples (Lin et al., 2021; Zhang et al., 2021a; Chen et al., 2024a; Tang & Liu, 2022; Gu et al., 2024). It aims to under no any label information divide all samples into distinct sets such that samples within the same set have relatively higher similarity while different sets are with significant differences, thereby discovering the latent pattern relations embedded inside samples (Liu et al., 2020; Huang et al., 2021; Ma et al., 2024a; Yu et al., 2024a; Pan & Kang, 2021; Li et al., 2025; Chen et al., 2024b; Yu et al., 2023b). To generate high-quality results for IMVC, recently a series of eye-catching algorithms have been carefully devised (Zhang et al., 2022; Wang et al., 2021c; Yu et al., 2024d; Yang et al., 2023a; Chen et al., 2023; Yu et al., 2024c). For instance, Li et al. (2024a) describe the relationship between existing samples and prototypes through an incomplete graph instead of the full pair-wise instance graph to improve the computational efficiency, and produce the uniform similarity under parameter free searching. Long et al. (2024) utilize the prototypes in potential feature subspace to do low-rank approximation for the view correlations, and preserve the consistencies between views by decreasing the rotation sensitivity in the embedded space. Rather than the distance-oriented

---

*Corresponding Authors.

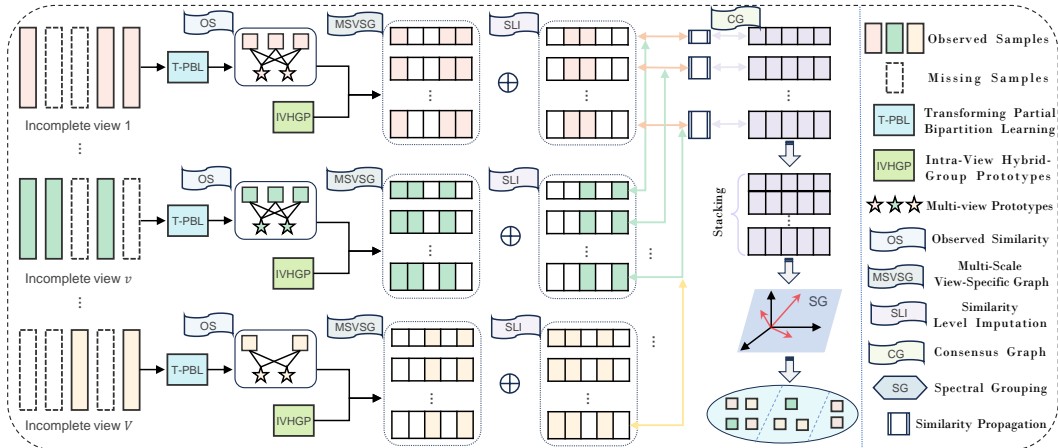

Figure 1: The overall framework of proposed SIIHPC. It firstly utilizes T-PBL to split out of observed similarity and then conducts SLI based on the connection between CG and view-specific similarities. Further, combined with IVHGP, it generates MSVSG to flexibly extract features on respective view.

weighting, He et al. (2023) construct an asymmetric matrix by structural prototype based metric learning to expand the late fusion, and accelerate the spectral generation through prototype inferred graph learning. Li et al. (2023) maintain the view versatility and instance commonality via a double stream learning framework, and use view-wise prototypes to exploit cluster-specific representations.

These approaches successfully achieve clustering result improvement from various perspectives, nevertheless, they typically choose to ignore the missing samples and only take advantages of observed samples to achieve the construction of bipartite similarity. This will miss out latent useful information from the missing samples, resulting in the generated similarity not that accurate. Also, due to the randomness of sample missing, the remaining observed samples are usually unpaired, which could lead to unbalanced cluster distribution and deteriorate the graph structure. Besides, they usually employ a single quantity of prototypes to extract all view information. This is apparently unreasonable since each view generally owns unique features, and a single quantity of prototypes could be not competent to adequately characterize all views, accordingly weakening the view information diversity.

To eliminate these issues, we present a simple yet effective IMVC method, SIIHPC, in this paper. The overall framework is described in Fig. 1. Concretely, we firstly transform partial bipartition learning under prototype orthogonality into the form containing original samples by utilizing the data reconstruction concept to split out of observed similarity, and then relax conventional non-negative constraints through a sample regularization skill to make the measure of similarity more free. Based on the criterion that one object appears on at least one view, we further introduce the learnable consensus graph, which is shared for all views, to provide unified structure. Afterwards, relying on the connection built between all view-specific bipartition similarities and the consensus graph, we gather the information from other views at the similarity level to assist imputing the incomplete parts of similarity on each view. On this basis, rather than a single prototype quantity for all views, we associate a group of hybrid prototype quantities for each individual view so that it can flexibly exploit features according to the characteristics of each view. Accordingly, the resultant graphs have various scales and in addition to balancing views, they also can more comprehensively characterize the overall similarity. In particular, we achieve these goals within one unified learning framework such that they are able to negotiate with each other towards the direction of mutual reinforcement. Then, to minimize the objective function, we adopt the alternate optimization idea and design an ingenious four-step solution scheme that cleverly solves the sub-problem through an auxiliary function with theoretically and experimentally proven monotonic-increasing properties. Subsequently, we stack all obtained multi-scale consensus graphs and perform spectral grouping action on the feature embedding that consists of the eigenvectors to generate the clustering results. After that, to demonstrate the effectiveness of SIIHPC, we organize experiments on multiple datasets and under different missing percentages. Numerous experiment results suggest that our SIIHPC has the ability to effectively cluster incomplete data. Main novelties in this paper are as follows:

- Unlike previous methods disregarding the missing samples when constructing bipartite similarity, this paper successfully imputes incomplete parts at the similarity level. Not only does this alleviate the adverse impacts caused by the unpairing of observed samples but also can take advantages of the potential useful information of missing samples to help characterize the similarity more accurately.

- Instead of a single prototype quantity for all views, this paper successfully generates a group of hybrid prototype quantities for each individual view to flexibly extract data features according to the characteristics of each view itself. The resulting graphs are with diverse scales and besides balancing views, they are also able to more comprehensively describe the overall similarity.

- To optimize the objective function, this paper carefully designs an alternate solving scheme, which decomposes the entire problem into four parts and solves the sub-problem via an ingenious auxiliary function with theoretically proven monotonic-increasing properties.

## 2 RELATED WORKS

As the information age progresses, multi-view data which commonly stems from diverse descriptions of the same instances is becoming increasingly widespread (Ma et al., 2024b; Peng et al., 2019; Kang et al., 2020b; Yu et al., 2023a; Qin et al., 2023; Zhang et al., 2024c; Yang et al., 2023b; Lu et al., 2024). Accordingly, clustering technology receives growing interest owing to its ability effectively grouping multi-view data without needing label information (Wan et al., 2024; Liang et al., 2024; Kang et al., 2020a; Lin et al., 2023b; Wang et al., 2021a; Huang et al., 2022; Yu et al., 2024b). However, due to factors like sensor breakdown or environment change, it is inevitable that some samples are missing/incomplete on certain views, causing traditional clustering methods not working properly and inducing the IMVC problem (Wang et al., 2022b; Zhang et al., 2019; Xu et al., 2024; Zeng et al., 2024; Lv et al., 2022). For effectively tackling this problem, many remarkable works have been proposed successively from various perspectives, such as (Zhao et al., 2023; Yang et al., 2024; Zhang et al., 2024a; Wang et al., 2021b; Huang et al., 2023; Xu et al., 2022; Zhang et al., 2024b).

Let matrices $\{\mathbf{D}_v \in \mathbb{R}^{d_v \times n}\}_{v=1}^V$ and vectors $\{r_v \in \mathbb{R}^{n_v}\}_{v=1}^V$ denote the overall data and indexes of observed data respectively, then, the basic IMVC framework can be expressed as

$$\min_{\mathbf{X}_v} \sum_{v=1}^V \|\mathbf{D}_v \mathbf{W}_v - \mathbf{H}_v \mathbf{X}_v \mathbf{W}_v\|_F^2 + \lambda \|\mathbf{X}_v\|_F^2 \quad s.t. \quad \mathbf{X}_v^\top \mathbf{1} = \mathbf{1}, \mathbf{X}_v \geq 0, \tag{1}$$

where the indicator matrix $\mathbf{W}_v \in \mathbb{R}^{n \times n_v}$ measures the incompleteness of view $v$, and its elements consist of $[\mathbf{W}_v]_{i,j} = 1 \ when \ [r_v]_j == i \ otherwise \ [\mathbf{W}_v]_{i,j} = 0, \ \forall j = 1, 2, \cdots, n_v$. $\mathbf{H}_v \in \mathbb{R}^{d_v \times m}$ denotes the prototype matrix on view $v$, and is intended to approximately characterize the data $\mathbf{D}_v$. $\mathbf{X}_v \in \mathbb{R}^{m \times n}$ denotes the incomplete similarity matrix on view $v$. The clustering results can be obtained by first fusing all learned $\mathbf{X}_v$ and then performing spectral embedding partitioning on it.

Following this paradigm, Chen et al. (2023) introduce tensor learning to exploit the low-rankness between views and utilize high-level view correlations captured by tensor to assist the learning of prototypes. Wang et al. (2022a) conduct a group of projectors to guarantee the dimension consistency of prototypes and aggregate different view information via an uniform fusion scheme. Xu et al. (2023a) regard the common features among views as prototypes and perform the distribution alignment by maximizing the mutual information between prototypes and view-wise features. Lin et al. (2023a) choose to concatenate view-representations as prototypes on each view and preserve the consistency by minimizing the distance between prototypes and within-cluster instances. Wen et al. (2021a) adopt the within-view maintenance and between-view inference strategy to decrease the adverse impact of information unbalance and encourage cluster structure directly reflected in representations. Lin et al. (2024) concurrently recoup and infer features in latent embedding space to explore the correlations between views and utilize an exploratory scheme to update all parameters. Xia et al. (2022) employ tensor norm to extract complementary information and introduce connectivity constraint to capture the spatial structure hidden into similarity. Xu et al. (2023b) design a two-branch, common and private, variable strategy to leverage representations and improve the robustness to senseless information via a controllable way. Wen et al. (2021b) devise a graph regularizer to maintain the local geometric similarities between views and utilize semantic coherence constraints to stimulate uniform features.

## 3 METHODOLOGY

Rethinking (1), its nature is to reconstruct $\mathbf{D}_v$ using $\mathbf{H}_v\mathbf{X}_v$ under given $\mathbf{H}_v$. Unlike the fixing strategy, we firstly make prototype learnable and then introduce orthogonal constraint to strengthen its discrimination, i.e., $\mathbf{H}_v^\top\mathbf{H}_v = \mathbf{I}$. On this basis, we have $\mathbf{H}_v^\top\mathbf{D}_v\mathbf{W}_v = \mathbf{X}_v\mathbf{W}_v$. Then, the observed parts can be splited out through $\mathbf{X}_v\mathbf{W}_v\mathbf{W}_v^\top = \mathbf{H}_v^\top\mathbf{D}_v\mathbf{W}_v\mathbf{W}_v^\top$. Notice that the item $\mathbf{H}_v^\top\mathbf{D}_v$ can be regarded as the cosine similarity between $\mathbf{H}_v^\top$ and $\mathbf{D}_v$ when all columns of $\mathbf{D}_v$ are unit vectors. Hence, we choose to do normalization on $\mathbf{D}_v$, which expands the similarity range from $[0, 1]$ to $[-1, 1]$, more freely measuring the similarity. Subsequently, we introduce a consensus graph $\mathbf{G}$ to aggregate information from different views, and impute the incomplete parts by utilizing $\mathbf{H}_v^\top\mathbf{D}_v\mathbf{W}_v\mathbf{W}_v^\top$ and $\mathbf{G}$. Further, to avoid a single prototype quantity for all views, we provide a group of hybrid prototype quantities $\{m_1, m_2, \cdots, m_s, \cdots, m_S\}$ for each view $v$ to flexibly extract features according to the characteristics of each view itself. Consequently, we have $\mathbf{X}_{v,s}\mathbf{W}_v\mathbf{W}_v^\top = \mathbf{H}_{v,s}^\top\mathbf{D}_v\mathbf{W}_v\mathbf{W}_v^\top$. Besides, to adaptively adjust the importance between prototype quantities, we associate a learnable weight, $a_{v,s}$, for each prototype quantity on each view. Finally, our SIIHPC can be formulated as

$$\min_{\mathbf{A},\mathbf{H}_{v,s},\mathbf{Q}_{v,s},\mathbf{G}_s} \sum_{v=1}^{V}\sum_{s=1}^{S} a_{v,s} \left( \left\| \mathbf{H}_{v,s}^\top\mathbf{D}_v\mathbf{W}_v\mathbf{W}_v^\top + \mathbf{Q}_{v,s}\mathbf{M}_v\mathbf{M}_v^\top - \mathbf{G}_s \right\|_F^2 + \lambda \left\| \mathbf{G}_s \right\|_F^2 \right) + \beta \left\| \mathbf{A} \right\|_F^2$$

$$s.t. \ \mathbf{H}_{v,s}^\top\mathbf{H}_{v,s} = \mathbf{I}_{m_s}, -1 \leq \mathbf{Q}_{v,s} \leq 1, -1 \leq \mathbf{G}_s \leq 1, \mathbf{A1} = \mathbf{1}, 0 \leq \mathbf{A}, \tag{2}$$

where $\mathbf{H}_{v,s} \in \mathbb{R}^{d_v \times m_s}$ denotes the prototype matrix with the $s$-th quantity on view $v$. $\mathbf{M}_v$ consists of $[\mathbf{M}_v]_{i,j} = 1 \ when \ [h_v]_j == i \ otherwise \ [\mathbf{M}_v]_{i,j} = 0, \ \forall j = 1, 2, \cdots, n - n_v; i = 1, 2, \cdots, n.$ $h_v = \{z | z \in T_a \ and \ z \notin T_o\}$ where $T_a = \{1, 2, \cdots, n\}$ and $T_o = \{[r_v]_1, [r_v]_2, \cdots, [r_v]_{n_v}\}$. $\mathbf{Q}_{v,s} \in \mathbb{R}^{m_s \times n}$ is the imputation matrix with the $s$-th scale on view $v$. $\mathbf{A} \in \mathbb{R}^{V \times S}$ consists of $a_{v,s}$.

## 4 OPTIMIZATION

Due to the non-convexity when jointly considering all variables in (2), we alternatively optimize each variable via the following four-step updating skill.

***Step 1:** Optimizing the Prototype Matrix $\mathbf{H}_{v,s}$*

Under fixed $\mathbf{Q}_{v,s}$, $\mathbf{G}_s$ and $\mathbf{A}$, we can simplify the optimization problem (2) as

$$\max_{\mathbf{H}_{v,s}} \mathrm{Tr}\left( \mathbf{H}_{v,s}^\top\widehat{\mathbf{L}}_v\mathbf{H}_{v,s} + \mathbf{H}_{v,s}^\top\mathbf{P}_{v,s} \right) \ \ s.t. \ \mathbf{H}_{v,s}^\top\mathbf{H}_{v,s} = \mathbf{I}_{m_s}, \tag{3}$$

where the matrix $\widehat{\mathbf{L}}_v = \varphi_v\mathbf{I}_{d_v} - \mathbf{L}_v$, the scalar $\varphi_v$ represents the largest eigenvalue of $\mathbf{L}_v$, $\mathbf{L}_v = \mathbf{D}_v\mathbf{W}_v\mathbf{W}_v^\top\mathbf{W}_v\mathbf{W}_v^\top\mathbf{D}_v^\top$, $\mathbf{P}_{v,s} = 2\mathbf{D}_v\mathbf{W}_v\mathbf{W}_v^\top \left( \mathbf{G}_s - \mathbf{Q}_{v,s}\mathbf{M}_v\mathbf{M}_v^\top \right)^\top$.

Denote the function $g(\mathbf{H}_{v,s}) = \mathrm{Tr}(\mathbf{H}_{v,s}^\top\widehat{\mathbf{L}}_v\mathbf{H}_{v,s} + \mathbf{H}_{v,s}^\top\mathbf{P}_{v,s})$, its derivative as $\nabla g((\mathbf{H}_{v,s}))$, the value of $\mathbf{H}_{v,s}$ at the $r$-th iteration as $(\mathbf{H}_{v,s})^r$, the singular value decomposition results of $\nabla g((\mathbf{H}_{v,s})^r)$ as $(\mathbf{U}_{v,s})^r (\mathbf{\Sigma}_{v,s})^r (\mathbf{V}_{v,s}^\top)^r$. Then, we have the following two lemmas hold.

**Lemma 1.** *Under $(\mathbf{H}_{v,s})^{r+1}$ taking $(\mathbf{U}_{v,s})^r (\mathbf{V}_{v,s}^\top)^r$, for the trace of $\mathbf{H}_{v,s}$ and its derivative, we have*

$$\mathrm{Tr}\left( \left[ (\mathbf{H}_{v,s}^\top)^{r+1} - (\mathbf{H}_{v,s}^\top)^r \right] \nabla g\left( (\mathbf{H}_{v,s})^r \right) \right) \geq 0, \tag{4}$$

*where $\nabla g\left( (\mathbf{H}_{v,s})^r \right)$ denotes the derivative value at the $r$-th iteration.*

**Lemma 2.** *For the trace of $\mathbf{H}_{v,s}$ at the $r$-th iteration and $(r+1)$-th iteration, we have*

$$\mathrm{Tr}\left( (\mathbf{H}_{v,s}^\top)^{r+1}\widehat{\mathbf{L}}_v \left[ (\mathbf{H}_{v,s})^{r+1} - (\mathbf{H}_{v,s})^r \right] \right) \geq \mathrm{Tr}\left( \left[ (\mathbf{H}_{v,s}^\top)^{r+1} - (\mathbf{H}_{v,s}^\top)^r \right] \widehat{\mathbf{L}}_v (\mathbf{H}_{v,s})^r \right). \tag{5}$$

In conjunction with **Lemma** 1 and **Lemma** 2, we have the following theorem holds.

**Theorem 1.** *For the function g, under any $(\mathbf{H}_{v,s})^r$ and $(\mathbf{H}_{v,s})^{r+1} = (\mathbf{U}_{v,s})^r (\mathbf{V}_{v,s}^\top)^r$, we have $g(\mathbf{H}_{v,s})$ is monotonically increasing.*

According to **Theorem** 1, we can determine $\mathbf{H}_{v,s}$ by comparing the objective value at current iteration and that at previous iteration. **Algorithm** 1 summaries the overall procedure of optimizing $\mathbf{H}_{v,s}$.

---

**Algorithm 1** The procedure of optimizing $\mathbf{H}_{v,s}$ in (3).

---

**Input:** The matrices $\mathbf{H}_{v,s}, \mathbf{Q}_{v,s}, \mathbf{G}_s, \mathbf{A}, \mathbf{D}_v, \mathbf{W}_v, \mathbf{M}_v$.
    Construct the function $g(\mathbf{H}_{v,s})$.
  1: **while** $g((\mathbf{H}_{v,s})^{r+1}) - g((\mathbf{H}_{v,s})^r)/g((\mathbf{H}_{v,s})^r) \leq 1e-3$ **do**
  2:    Compute the derivative function $\nabla g\left((\mathbf{H}_{v,s})^r\right)$.
  3:    Generate the singular matrices $(\mathbf{U}_{v,s})^r$ and $\left(\mathbf{V}_{v,s}^\top\right)^r$.
  4:    Assign $(\mathbf{H}_{v,s})^{r+1}$ by $(\mathbf{H}_{v,s})^{r+1} = (\mathbf{U}_{v,s})^r \left(\mathbf{V}_{v,s}^\top\right)^r$.
  5:    $r = r+1$.
  6: **end while**
**Output:** The prototype matrices $\{\mathbf{H}_{v,s}\}_{v=1,s=1}^{V,\ S}$.

---

**Remark 1.** *Due to $\mathbf{W}_v\mathbf{W}_v^\top \in \mathbb{R}^{n \times n}$ and $\mathbf{M}_v\mathbf{M}_v^\top \in \mathbb{R}^{n \times n}$, calculating $\mathbf{L}_v$ and $\mathbf{P}_{v,s}$ needs at least $\mathcal{O}(n^2)$ computing overhead. Noticed that $\mathbf{W}_v\mathbf{W}_v^\top$ and $\mathbf{M}_v\mathbf{M}_v^\top$ are diagonal matrices with elements 0 or 1, by virtue of Hadamard product, we can transform $\mathbf{L}_v$ and $\mathbf{P}_{v,s}$ as $\mathbf{D}_v \odot \mathbf{B}_v \cdot \mathbf{B}_v^\top \odot \mathbf{D}_v^\top$ and $2\mathbf{D}_v \odot \mathbf{B}_v(\mathbf{G}_s - \mathbf{Q}_{v,s} \odot \mathbf{C}_v)^\top$, where $\mathbf{B}_v = \mathbf{1}_{d_v} \cdot [\sum_{j=1}^{n_v}[\mathbf{W}_v]_{1,j}, \sum_{j=1}^{n_v}[\mathbf{W}_v]_{2,j}, \cdots, \sum_{j=1}^{n_v}[\mathbf{W}_v]_{n,j}]$ and $\mathbf{C}_v = \mathbf{1}_{m_s} \cdot [\sum_{j=1}^{n-n_v}[\mathbf{M}_v]_{1,j}, \sum_{j=1}^{n-n_v}[\mathbf{M}_v]_{2,j}, \cdots, \sum_{j=1}^{n-n_v}[\mathbf{M}_v]_{n,j}]$. After transforming, the computing complexity is reduced to $\mathcal{O}(n)$.*

**Step 2:** *Optimizing the Similarity Imputation Matrix $\mathbf{Q}_{v,s}$*

Under fixed $\mathbf{H}_{v,s}$, $\mathbf{G}_s$ and $\mathbf{A}$, we can simplify the problem (2) as

$$\min_{\mathbf{Q}_{v,s}} \left\|\mathbf{Q}_{v,s}\mathbf{M}_v\mathbf{M}_v^\top - \mathbf{J}_{v,s}\right\|_F^2 \quad s.t. \ -1 \leq \mathbf{Q}_{v,s} \leq 1, \tag{6}$$

where the matrix $\mathbf{J}_{v,s} = \mathbf{G}_s - \mathbf{H}_{v,s}^\top\mathbf{D}_v\mathbf{W}_v\mathbf{W}_v^\top$. $\mathbf{M}_v$ is an indicator matrix, and therefore we can determine $\mathbf{Q}_{v,s}$ by taking the value of corresponding index of $\mathbf{J}_{v,s}$. To guarantee the feasible region, we can regularize the solution by first comparing it and $\pm 1$ and then performing truncation operation.

**Remark 2.** *Due to the direct assignment operation, the computing overhead of optimizing $\mathbf{Q}_{v,s}$ is mainly from the construction of $\mathbf{J}_{v,s}$. Inspired by **Remark 1**, the item $\mathbf{H}_{v,s}^\top\mathbf{D}_v\mathbf{W}_v\mathbf{W}_v^\top$ can be transformed as $\mathbf{H}_{v,s}^\top\mathbf{D}_v \odot \mathbf{B}_v$. Therefore, the computing complexity about optimizing $\mathbf{Q}_{v,s}$ is $\mathcal{O}(n)$.*

**Step 3:** *Optimizing the Unified Representation Matrix $\mathbf{G}_s$*

Under fixed $\mathbf{H}_{v,s}$, $\mathbf{Q}_{v,s}$ and $\mathbf{A}$, we can simplify the problem (2) as

$$\min_{\mathbf{G}_s} \text{Tr}\left(\mathbf{G}_s^\top\left(\sum_{v=1}^V a_{v,s}(1+\lambda)\mathbf{I}_{m_s}\right)\mathbf{G}_s - 2\left(\sum_{v=1}^V (a_{v,s}\mathbf{F}_{v,s})^\top\right)\mathbf{G}_s\right), \quad s.t. \ -1 \leq \mathbf{G}_s \leq 1, \tag{7}$$

where the matrix $\mathbf{F}_{v,s} = \mathbf{H}_{v,s}^\top\mathbf{D}_v\mathbf{W}_v\mathbf{W}_v^\top + \mathbf{Q}_{v,s}\mathbf{M}_v\mathbf{M}_v^\top$. After expanding the trace by elements, we can equivalently transform the objective as

$$\min_{[\mathbf{G}_s]_{:,j}} [\mathbf{G}_s]_{:,j}^\top\left(\sum_{v=1}^V a_{v,s}(1+\lambda)\mathbf{I}_{m_s}\right)[\mathbf{G}_s]_{:,j} - 2\left(\sum_{v=1}^V a_{v,s}[\mathbf{F}_{v,s}]_{:,j}\right)^\top[\mathbf{G}_s]_{:,j}. \tag{8}$$

For the feasible region, we can split it into $-1 \leq [\mathbf{G}_s]_{:,j} \leq 1, j = 1, 2, \cdots, n$. Therefore, the problem (7) is transformed as a quadratic programming (QP) problem, and can be effectively solved using existing software packages.

**Remark 3.** *Solving each column vector $[\mathbf{G}_s]_{:,j}$ by QP consumes $\mathcal{O}(m_s^3)$ overhead. Therefore, the computing overhead of optimizing $\mathbf{G}_s$ is $\mathcal{O}(m_s^3 n)$. Note that the number of prototypes $m_s$ is not related to $n$ and usually is far less than $n$, accordingly, the computing complexity of $\mathbf{G}_s$ is also $\mathcal{O}(n)$.*

**Step 4:** *Optimizing the Prototype Balance Matrix $\mathbf{A}$*

Under fixed $\mathbf{H}_{v,s}$, $\mathbf{Q}_{v,s}$ and $\mathbf{G}_s$, we can simplify the problem (2) as

$$\min_{\mathbf{A}} \sum_{v=1}^V \sum_{s=1}^S a_{v,s}p_{v,s} + \beta\|\mathbf{A}\|_F^2 \quad s.t. \ \mathbf{A}\mathbf{1} = \mathbf{1}, 0 \leq \mathbf{A}, \tag{9}$$

---

**Algorithm 2** The procedure of solving the problem (2).

---

**Input:** Data matrix $\mathbf{D}_v$, index vectors $b_v$, hyper-parameters $\lambda$ and $\beta$, $v = 1, 2, \cdots, V$.

  Construct indicator matrices $\mathbf{W}_v$ and $\mathbf{M}_v$.

 1: **while** $(f_{obj}(t) - f_{obj}(t+1))/f_{obj}(t) <= 1e - 4$ **do**

 2:  Optimize the variable $\mathbf{H}_{v,s}$ by **Algorithm** 1.

 3:  Optimize the variable $\mathbf{Q}_{v,s}$ by solving (6).

 4:  Optimize the variable $\mathbf{G}_s$ by solving (7).

 5:  Optimize the variable $\mathbf{A}$ by (11).

 6: **end while**

**Output:** The unified representation matrices $\{\mathbf{G}_s\}_{s=1}^S$.

---

where $p_{v,s} = \left\| \mathbf{H}_{v,s}^\top \mathbf{D}_v \mathbf{W}_v \mathbf{W}_v^\top + \mathbf{Q}_{v,s} \mathbf{M}_v \mathbf{M}_v^\top - \mathbf{G}_s \right\|_F^2 + \lambda \left\| \mathbf{G}_s \right\|_F^2$. Due to the constraints being for the row of $\mathbf{A}$, we can further transform the problem (9) as

$$\min_{a_{v,s}} \sum_{s=1}^{S} a_{v,s} p_{v,s} + \beta a_{v,s}^2 \quad s.t. \ \mathbf{a}_{v,:} \mathbf{1} = 1, 0 \le \mathbf{a}_{v,:}, \tag{10}$$

where the vector $\mathbf{a}_{v,:}$ denotes the $v$-th row of $\mathbf{A}$. For the above optimization problem, we can get its closed-form solution as

$$\mathbf{a}_{v,:} = \left( \frac{\frac{\mathbf{p}_{v,:} \cdot \mathbf{1} \cdot \mathbf{1}^\top}{2\beta} + \mathbf{1}^\top}{S} - \frac{\mathbf{p}_{v,:}}{2\beta} \right)_+, \tag{11}$$

where the vector $\mathbf{p}_{v,:}$ is composed of $p_{v,s}, s = 1, 2, \cdots, S$.

**Remark 4.** *Owing to the closed-form solution, the computing overhead is mainly from the construction of $\mathbf{p}_{v,:}$. According to **Remark 1**, we can obtain that constructing each $p_{v,s}$ takes $\mathcal{O}(n)$. Accordingly, solving $\mathbf{a}_{v,:}$ takes $\mathcal{O}(Sn)$. Solving the overall $\mathbf{A}$ will take $\mathcal{O}(VSn)$, which is also $\mathcal{O}(n)$.*

We summary the overall procedure for solving the optimization problem (2) in **Algorithm** 2, where $f_{obj}(t)$ denotes the objective value at the $t$-th iteration.

After obtaining the unified representation matrices $\{\mathbf{G}_s\}_{s=1}^S$, we concatenate them by row and subsequently perform spectral grouping on it to generate the data clustering results.

**Remark 5.** *The overall computing complexity of **Algorithm** 2 is $\mathcal{O}(n)$ since updating $\mathbf{H}_{v,s}$, $\mathbf{Q}_{v,s}$, $\mathbf{G}_s$ and $\mathbf{A}$ all take $\mathcal{O}(n)$, which consequently enables it to be expanded to large-scale tasks.*

**Remark 6.** *Storing the optimization variables $\mathbf{H}_{v,s}$, $\mathbf{Q}_{v,s}$, $\mathbf{G}_s$ and $\mathbf{A}$ takes $\mathcal{O}(m_s d_v)$, $\mathcal{O}(m_s n)$, $\mathcal{O}(m_s n)$ and $\mathcal{O}(SV)$, respectively. Therefore, the space complexity of **Algorithm** 2 is also $\mathcal{O}(n)$.*

## 5 EXPERIMENTS

### 5.1 BASELINES AND DATASETS

We conduct all experiments on six public multi-view datasets, and their details are presented in Table 1, where SS: Sample Size, NV: Number of Views, FD: Feature Dimension, NC: Number of Clusters.

Table 1: Dataset Description

| Dataset | SS | NV | FD | NC |
|---------|-----|-----|--------------------|-----|
| BDGPFEA | 2500 | 3 | 1000/500/250 | 5 |
| NUSOBJECT | 6251 | 5 | 129/74/145/226/65 | 10 |
| VGGFACEFIFTY | 16936 | 4 | 944/576/512/640 | 50 |
| VGGFACEHUND | 36287 | 4 | 512/576/640/944 | 100 |
| YOUTUBEFACE | 63896 | 4 | 640/944/576/512 | 20 |
| FASHMINST | 70000 | 4 | 576/512/944/640 | 10 |

The following methods are used as baselines in this paper to illustrate the effectiveness of SIIHPC:

Localized Sparsity (**LSIMVC** (Liu et al., 2023)), Refined Graph Structure (**GSRIMC** (Li et al., 2024b)), High-Order Correlation (**HCPIMSC** (Li et al., 2022)), Efficient Effective Regularizer (**EEIMVC** (Liu et al., 2021)), Low-Rank Graph (**LRGRIMVC** (Cui et al., 2024)), Consensus Bipartite Graph (**IMVCCBG** (Wang et al., 2022a)), Balance Guidance (**BGIMVSC** (Sun et al., 2023)), Late Fusion (**OSLFIMVC** (Zhang et al., 2021b)), Neighbor Group Structure (**NGSPCGL** (Wong et al., 2023)), Projections (**PIMVC** (Deng et al., 2023)), Parameter-Free Scalable Prototype Graph (**PSIMVC** (Li et al., 2024a)), Structured Anchor-Inferred Graph (**SAGL** (He et al., 2023)), Local Structure Consensus Graph (**HCLSCGL** Wen et al. (2023)).

## 5.2 RESULTS AND DISCUSSIONS

Table 2: Clustering Results on Benchmark Datasets

| Method | BDGPFEA 30% | | | 50% | | | 70% | | | NUSOBJECT 30% | | | 50% | | | 70% | | |
|---|---|---|---|---|---|---|---|---|---|---|---|---|---|---|---|---|---|---|
| | ACC | NMI | PUR | ACC | NMI | PUR | ACC | NMI | PUR | ACC | NMI | PUR | ACC | NMI | PUR | ACC | NMI | PUR |
| LSIMVC | 26.56 | 6.14 | 26.56 | 26.81 | 7.18 | 26.56 | 27.40 | 8.35 | 28.07 | 21.70 | 9.03 | 31.23 | 21.81 | 8.43 | 30.99 | 20.57 | 7.43 | 30.52 |
| GSRIMC | 39.84 | 14.22 | 39.96 | 38.30 | 13.15 | 36.78 | 34.41 | 11.22 | 34.17 | 23.03 | 8.34 | 32.73 | 20.91 | 7.58 | 32.04 | 20.46 | 7.86 | 31.20 |
| HCPIMSC | 34.12 | 12.65 | 36.36 | 32.20 | 12.41 | 35.24 | 33.16 | **11.81** | 34.58 | 21.56 | 6.38 | 29.31 | 21.33 | 8.93 | 30.18 | **22.87** | 8.07 | 29.81 |
| EEIMVC | 35.70 | 14.47 | 36.34 | 33.23 | 12.39 | 36.05 | 31.02 | 10.37 | 33.64 | 21.51 | 6.17 | 13.51 | 21.07 | 8.97 | 13.42 | 20.13 | 8.14 | 13.17 |
| LRGRIMVC | 34.71 | 12.58 | 35.74 | 31.43 | 9.12 | 32.63 | 27.25 | 4.80 | 27.27 | 21.43 | 7.25 | 30.17 | **22.62** | 7.83 | 30.99 | 21.73 | 8.16 | 30.15 |
| IMVCCBG | **40.05** | 15.01 | **40.17** | 38.10 | 11.49 | 36.76 | 34.78 | 10.03 | 34.06 | 22.59 | 8.36 | 31.96 | 21.35 | 7.90 | 32.85 | 22.05 | 7.43 | 31.16 |
| BGIMVSC | 22.65 | 3.19 | 23.26 | 26.88 | 9.68 | 27.08 | 24.04 | 4.72 | 24.56 | 19.06 | 0.30 | 22.86 | 19.13 | 0.34 | 22.91 | 19.06 | 0.31 | 22.86 |
| OSLFIMVC | 30.38 | 9.58 | 36.47 | 31.84 | 9.12 | 35.37 | 31.68 | 8.80 | 35.73 | 21.88 | 7.67 | 32.34 | 20.68 | 6.91 | 33.00 | 18.41 | 4.63 | 29.58 |
| NGSPCGL | 29.95 | 6.64 | 31.13 | 29.54 | 6.07 | 29.89 | 27.15 | 5.70 | 27.64 | 23.09 | 7.83 | 30.78 | 19.97 | 4.62 | 28.32 | 17.47 | 2.22 | 25.20 |
| PIMVC | 34.03 | 14.62 | 35.99 | 33.04 | 12.92 | 33.54 | 34.41 | 11.60 | 35.41 | 21.29 | **9.36** | 31.07 | 21.08 | 8.52 | 31.12 | 19.44 | 7.89 | 31.36 |
| PSIMVC | 34.00 | 12.51 | 35.58 | 31.98 | 9.55 | 33.65 | 30.12 | 9.15 | 32.45 | 19.65 | 8.25 | 29.11 | 20.09 | 8.91 | 30.21 | 22.07 | 7.91 | 30.35 |
| SAGL | 23.76 | 1.69 | 23.92 | 23.03 | 1.41 | 23.60 | 28.52 | 4.09 | 29.56 | 20.48 | 7.67 | 27.46 | 20.39 | 6.91 | 26.16 | 18.39 | 6.63 | 26.47 |
| HCLSCGL | 29.80 | 7.12 | 31.40 | 24.28 | 3.12 | 25.08 | 28.25 | 4.31 | 28.55 | 21.93 | 7.54 | 30.68 | 21.59 | 7.79 | 31.49 | 20.28 | 7.81 | 31.31 |
| Ours | 38.80 | **15.21** | 39.97 | **40.31** | 13.88 | **40.31** | **35.04** | 11.54 | **37.30** | **23.30** | 9.14 | **32.87** | 22.38 | **9.21** | **33.92** | 21.46 | **8.36** | 31.49 |

| Method | VGGFACEFIFTY 30% | | | 50% | | | 70% | | | VGGFACEHUND 30% | | | 50% | | | 70% | | |
|---|---|---|---|---|---|---|---|---|---|---|---|---|---|---|---|---|---|---|
| LSIMVC | 8.45 | 10.78 | 8.65 | 7.30 | 9.83 | 7.56 | 6.94 | 9.08 | 7.18 | | | | | | | | | |
| GSRIMC | | | | N/A | | | | | | | | | N/A | | | | | |
| HCPIMSC | 10.33 | 12.36 | 12.14 | 10.54 | 11.90 | 10.85 | 10.85 | 9.23 | 10.16 | | | | | | | | | |
| EEIMVC | 6.05 | 14.03 | 5.94 | 5.60 | 14.15 | 5.50 | 5.33 | 13.29 | 5.23 | 3.37 | 7.32 | 4.78 | 3.41 | 6.89 | 5.67 | 3.20 | 6.27 | 5.74 |
| LRGRIMVC | 9.21 | 13.23 | 11.37 | 10.02 | 12.48 | 11.45 | 9.15 | 11.58 | **12.56** | | | | N/A | | | | | |
| IMVCCBG | 12.13 | 14.25 | 13.11 | 11.52 | 13.29 | 12.40 | 10.80 | 12.35 | 11.66 | 8.12 | 14.23 | 8.92 | 7.52 | 13.25 | 8.25 | 6.80 | 12.20 | 7.06 |
| BGIMVSC | 6.49 | 9.83 | 6.83 | 7.34 | 9.45 | 7.19 | 6.76 | 9.85 | 7.19 | | | | N/A | | | | | |
| OSLFIMVC | 8.50 | 8.79 | 8.96 | 6.98 | 6.70 | 7.58 | 6.01 | 5.09 | 6.60 | 5.54 | 9.59 | 5.97 | 4.62 | 7.54 | 5.05 | 3.60 | 5.81 | 4.08 |
| NGSPCGL | 6.50 | 6.47 | 7.19 | 6.24 | 6.54 | 6.74 | 6.08 | 6.24 | 6.75 | | | | N/A | | | | | |
| PIMVC | 9.40 | 13.36 | 11.07 | 9.06 | 12.52 | 11.12 | 8.78 | 11.89 | 12.06 | 6.10 | 13.42 | 7.32 | 5.97 | 12.91 | 7.11 | 5.68 | 12.36 | 6.72 |
| PSIMVC | 10.63 | 12.50 | 11.58 | 9.54 | 11.33 | 10.49 | 9.06 | 10.45 | 9.92 | 6.17 | 11.04 | 6.71 | 5.28 | 10.58 | 5.89 | 5.51 | 9.91 | 6.04 |
| SAGL | 8.25 | 9.33 | 9.75 | 6.54 | 9.65 | 6.75 | 5.84 | 9.65 | 8.85 | 5.84 | 10.54 | 6.36 | 4.85 | 10.13 | 4.74 | 3.84 | 9.32 | 4.54 |
| HCLSCGL | 5.65 | 9.55 | 5.74 | 4.18 | 8.68 | 4.62 | 4.67 | 8.55 | 5.01 | 3.05 | 10.32 | 4.26 | 3.05 | 10.12 | 4.13 | 3.15 | 9.51 | 4.02 |
| Ours | **12.52** | **14.91** | **13.44** | **12.31** | **14.48** | **13.21** | **11.18** | **13.35** | 11.96 | **8.26** | **14.94** | **9.13** | **7.55** | **13.85** | **8.39** | **6.82** | **12.69** | **7.55** |

| Method | YOUTUBEFACE 30% | | | 50% | | | 70% | | | FASHMINST 30% | | | 50% | | | 70% | | |
|---|---|---|---|---|---|---|---|---|---|---|---|---|---|---|---|---|---|---|
| LSIMVC | | | | | | | | | | | | | | | | | | |
| GSRIMC | | | | | | | | | | | | | | | | | | |
| HCPIMSC | | | | N/A | | | | | | | | | N/A | | | | | |
| EEIMVC | | | | | | | | | | | | | | | | | | |
| LRGRIMVC | | | | | | | | | | | | | | | | | | |
| IMVCCBG | 74.88 | 79.28 | 77.29 | 72.39 | 78.33 | 77.04 | 70.61 | 78.60 | **77.97** | 58.18 | 57.58 | 62.22 | 58.01 | 58.01 | 61.75 | 56.27 | 56.48 | 60.17 |
| BGIMVSC | | | | N/A | | | | | | | | | N/A | | | | | |
| OSLFIMVC | 61.85 | 70.25 | 69.84 | 60.78 | 69.27 | 69.56 | 61.99 | 68.27 | 67.46 | 41.73 | 36.25 | 47.28 | 41.67 | 34.36 | 45.37 | 41.23 | 32.89 | 44.76 |
| NGSPCGL | | | | N/A | | | | | | | | | N/A | | | | | |
| PIMVC | | | | | | | | | | | | | | | | | | |
| PSIMVC | 68.10 | 75.67 | 74.33 | 67.48 | 72.29 | 74.49 | 67.67 | 71.64 | 74.17 | 50.11 | 52.23 | 51.92 | 54.09 | 57.19 | 58.11 | 54.42 | 54.78 | 57.73 |
| SAGL | 63.48 | 72.89 | 72.45 | 62.47 | 73.63 | 73.04 | 62.65 | 72.75 | 73.68 | 43.47 | 53.56 | 53.68 | 43.35 | 57.88 | 55.73 | 43.58 | 53.62 | 54.86 |
| HCLSCGL | | | | N/A | | | | | | | | | N/A | | | | | |
| Ours | **76.29** | **82.27** | **80.81** | **72.60** | **79.60** | **77.65** | **71.05** | **79.19** | 76.75 | **61.24** | **59.52** | **62.69** | **62.51** | **60.22** | **64.64** | **60.59** | **58.77** | **63.18** |

Table 2 presents the clustering results under multiple missing ratios (30%, 50%, 70%). We can get:

- The proposed SIIHPC receives preferable results than many comparison algorithms under multiple missing ratios and metrics. For instance, on the datasets VGGFACEHUND and FASHMINST, SIIHPC is consistently the best; on VGGFACEFIFTY and YOUTUBEFACE, SIIHPC obtains only two sub-optimal results totally; on BDGPFEA and NUSOBJECT, SIIHPC also makes desirable results. Therefore, SIIHPC can effectively tackle IMVC tasks.

- LSIMVC, GSRIMC, HCPIMSC, EEIMVC, LRGRIMVC, BGIMVSC, NGSPCGL, PIMVC and HCLSCGL can not normally run on slightly-larger dataset VGGFACEFIFTY, VG-GFACEHUND, YOUTUBEFACE or FASHMNIST, while the proposed SIIHPC is not only able to work properly under these circumstances but also makes favorable results. Therefore, SIIHPC is with relatively stronger practicality.

## 5.3 TIME AND MEMORY OVERHEAD COMPARISON

For illustrating SIIHPC's friendliness to computing resource and storage resource, we compare the running time (min) and memory overhead (GB) between all previously-mentioned IMVC methods, as reported in Table 3. According to this table, we can observe that:

- SIIHPC consumes fewer resources against most methods. For example, on datasets YOUTUBEFACE and FASHMINST, the algorithms OSLFIMVC and SAGL require 126.28GB, 82.96GB, 123.42GB and 99.01GB memory respectively while our SIIHPC only needs 6.01GB and 5.13GB. In other situations, the time overhead and memory overhead of SIIHPC are still relatively small. Therefore, SIIHPC is resource-friendly.

- In some cases, PIMVC, PSIMVC and IMVCCBG take lower running time and/or memory overhead, possibly because PIMVC learns representation in a common low-dimensional space instead of in diverse original space, PSIMVC employs only one bipartite graph to characterize the correlation between all views, and IMVCCBG utilizes the landmarks with a single dimension to extract features. Despite resource-saving, they generally can not integrate information from missing samples, accordingly giving inferior clustering results.

Table 3: Running Time and Memory Overhead Comparison

| Method | BDGPFEA | | NUSOBJECT | | VGGFACEFIFTY | | VGGFACEHUND | | YOUTUBEFACE | | FASHMINST | |
|---|---|---|---|---|---|---|---|---|---|---|---|---|
| | Time | Memo | Time | Memo | Time | Memo | Time | Memo | Time | Memo | Time | Memo |
| LSIMVC | 0.03 | 0.32 | 0.13 | 2.42 | 0.55 | 15.18 | | | | | | |
| GSRIMC | 2.10 | 2.82 | 28.88 | 28.07 | N/A | | N/A | | | | | |
| HCPIMSC | 2.79 | 1.93 | 43.33 | 17.25 | 741.45 | 104.84 | | | N/A | | N/A | |
| EEIMVC | 0.02 | 0.80 | 0.33 | 5.06 | 2.99 | 30.97 | 238.01 | 98.84 | | | | |
| LRGRIMVC | 3.35 | 1.11 | 60.59 | 10.09 | 492.37 | 61.87 | N/A | | | | | |
| IMVCCBG | 0.02 | 0.17 | 0.05 | 0.20 | 0.56 | 1.73 | 2.46 | **3.96** | **1.23** | 6.16 | **1.27** | 6.41 |
| BGIMVSC | 4.85 | 1.27 | 15.76 | 8.75 | 132.67 | 135.70 | N/A | | N/A | | N/A | |
| OSLFIMVC | 0.09 | 0.30 | 0.18 | 1.51 | 3.23 | 10.02 | 20.00 | 41.96 | 13.54 | 126.28 | 12.27 | 123.42 |
| NGSPCGL | 0.97 | 1.79 | 11.42 | 13.43 | 111.29 | 89.09 | N/A | | N/A | | N/A | |
| PIMVC | **0.01** | 0.46 | 0.40 | 2.53 | **0.36** | 17.20 | 3.37 | 76.58 | N/A | | N/A | |
| PSIMVC | 0.02 | 0.16 | **0.04** | **0.15** | 0.40 | **1.62** | **1.30** | 5.22 | 1.57 | 6.42 | 1.95 | 6.41 |
| SAGL | 0.22 | 0.40 | 0.41 | 2.22 | 7.54 | 16.29 | 45.06 | 26.96 | 38.39 | 82.96 | 30.07 | 99.01 |
| HCLSCGL | 0.17 | 1.84 | 4.14 | 13.92 | 309.87 | 91.51 | 4838.55 | 133.20 | N/A | | N/A | |
| Ours | 0.06 | **0.14** | 0.11 | 0.22 | 1.86 | 2.84 | 6.22 | 10.32 | 3.31 | **6.01** | 3.57 | **5.13** |

## 5.4 ABLATION

We utilize the similarity level imputation (SLI) to capture the latent useful information from missing samples and thereby improve the clustering performance. To verify its effectiveness, we do result comparison under these two situations, as shown in Table 4, where AB: Ablation, MR: Missing Ratio, NSLI: No-SLI. As seen, SLI results are always preferable. Therefore, our SLI scheme is functional.

Table 4: Similarity-level Imputation Effectiveness

| AB | MR | BDGPFEA | | | NUSOBJECT | | | VGGFACEFIFTY | | | VGGFACEHUND | | | YOUTUBEFACE | | | FASHMINST | | |
|---|---|---|---|---|---|---|---|---|---|---|---|---|---|---|---|---|---|---|---|
| | | ACC | NMI | PUR | ACC | NMI | PUR | ACC | NMI | PUR | ACC | NMI | PUR | ACC | NMI | PUR | ACC | NMI | PUR |
| NSLI | 30% | 28.15 | 3.83 | 30.38 | 22.60 | 7.29 | 31.85 | 6.71 | 7.22 | 7.51 | 4.83 | 9.93 | 5.53 | 46.19 | 40.95 | 51.69 | 46.99 | 33.74 | 49.41 |
| SLI | | **38.80** | **15.21** | **39.97** | **23.30** | **9.14** | **32.87** | **12.52** | **14.91** | **13.44** | **8.26** | **14.94** | **9.13** | **76.29** | **82.27** | **80.81** | **61.24** | **59.52** | **62.69** |
| NSLI | 50% | 29.74 | 3.97 | 30.98 | 21.09 | 6.20 | 31.10 | 5.33 | 4.45 | 6.00 | 3.73 | 6.69 | 4.26 | 26.07 | 16.16 | 28.60 | 37.85 | 24.03 | 40.85 |
| SLI | | **40.31** | **13.88** | **40.31** | **22.38** | **9.21** | **33.92** | **12.31** | **14.48** | **13.21** | **7.55** | **13.85** | **8.39** | **72.60** | **79.60** | **77.65** | **62.51** | **60.22** | **64.64** |
| NSLI | 70% | 26.39 | 1.68 | 26.80 | 18.05 | 3.25 | 27.86 | 5.01 | 3.76 | 5.64 | 3.12 | 4.98 | 3.56 | 15.82 | 15.40 | 17.46 | 25.15 | 9.26 | 27.31 |
| SLI | | **35.04** | **11.54** | **37.30** | **21.46** | **8.36** | **31.49** | **11.18** | **13.35** | **11.96** | **6.82** | **12.69** | **7.55** | **71.05** | **79.19** | **76.75** | **60.59** | **58.77** | **63.18** |

Unlike a single prototype quantity (SPQ) for all views, we introduce a group of hybrid prototype quantities (HPQ) $[1k, 2k, \cdots, 5k]$ for each view to flexibly exploit features where $k$ is the number of clusters. The ablation results are shown in Table 5 where PQ: prototype quantity. It can be seen that the HPQ results are consistently superior to any SPQ ones. Therefore, our HPQ scheme is effective.

Further, we adaptively adjust the importance of each prototype quantity via a learnable weight to more flexibly extract features. To verify its effectiveness, we present the ablation results in Table 6,

Table 5: Hybrid-group Prototype Quantity Effectiveness

| AB | MR | PQ | BDGPFEA | | | NUSOBJECT | | | VGGFACEFIFTY | | | VGGFACEHUND | | | YOUTUBEFACE | | | FASHMINST | | |
|---|---|---|---|---|---|---|---|---|---|---|---|---|---|---|---|---|---|---|---|---|
| | | | ACC | NMI | PUR | ACC | NMI | PUR | ACC | NMI | PUR | ACC | NMI | PUR | ACC | NMI | PUR | ACC | NMI | PUR |
| SPQ | 30% | m=1k | 29.77 | 6.07 | 30.45 | 12.75 | 1.30 | 23.03 | 9.48 | 11.21 | 10.45 | 5.15 | 10.63 | 6.03 | 65.28 | 72.52 | 70.51 | 53.16 | 56.70 | 58.30 |
| | | m=2k | 34.46 | 9.79 | 35.89 | 13.93 | 2.08 | 23.22 | 10.42 | 12.57 | 11.23 | 5.67 | 11.54 | 6.59 | 69.74 | 78.24 | 75.29 | 55.01 | 58.94 | 59.26 |
| | | m=3k | 28.71 | 6.70 | 30.60 | 15.15 | 3.64 | 24.36 | 11.29 | 13.35 | 12.06 | 6.41 | 12.27 | 7.27 | 70.68 | 80.63 | 76.57 | 53.50 | 58.61 | 57.34 |
| | | m=4k | 34.43 | 11.46 | 35.66 | 16.88 | 4.56 | 27.12 | 11.18 | 13.60 | 12.04 | 6.48 | 12.80 | 7.36 | 68.36 | 78.85 | 74.42 | 56.94 | 58.71 | 58.40 |
| | | m=5k | 34.84 | 9.74 | 36.15 | 17.46 | 4.21 | 27.17 | 12.00 | 14.03 | 12.92 | 7.06 | 12.56 | 7.44 | 69.81 | 76.88 | 75.33 | 56.12 | 55.80 | 59.85 |
| HPQ | | m=ours | **38.80** | **15.21** | **39.97** | **23.30** | **9.14** | **32.87** | **12.52** | **14.91** | **13.44** | **8.26** | **14.94** | **9.13** | **76.29** | **82.27** | **80.81** | **61.24** | **59.52** | **62.69** |
| SPQ | 50% | m=1k | 24.23 | 2.30 | 25.18 | 14.37 | 1.57 | 24.79 | 9.20 | 10.70 | 10.08 | 5.98 | 10.78 | 6.16 | 62.12 | 66.35 | 66.63 | 53.41 | 56.05 | 54.69 |
| | | m=2k | 34.87 | 7.83 | 34.87 | 18.88 | 5.68 | 29.75 | 10.77 | 12.81 | 11.71 | 6.12 | 11.31 | 6.32 | 66.96 | 74.15 | 71.43 | 50.97 | 55.58 | 54.41 |
| | | m=3k | 34.59 | 8.96 | 34.59 | 19.27 | 6.21 | 29.56 | 11.17 | 13.31 | 11.93 | 6.61 | 11.68 | 6.76 | 67.03 | 73.99 | 72.04 | 55.62 | 58.39 | 57.25 |
| | | m=4k | 34.98 | 9.44 | 35.74 | 20.13 | 7.01 | 30.80 | 11.24 | 13.51 | 12.17 | 6.54 | 11.74 | 6.68 | 70.21 | 77.23 | 74.64 | 58.47 | 58.69 | 59.24 |
| | | m=5k | 34.25 | 9.22 | 36.55 | 19.04 | 6.01 | 31.08 | 11.33 | 13.25 | 12.16 | 6.68 | 11.89 | 7.43 | 70.53 | 76.28 | 75.53 | 61.01 | 58.45 | 62.83 |
| HPQ | | m=ours | **40.31** | **13.88** | **40.31** | **22.38** | **9.21** | **33.92** | **12.31** | **14.48** | **13.21** | **7.55** | **13.85** | **8.39** | **72.60** | **79.60** | **77.65** | **62.51** | **60.22** | **64.64** |
| SPQ | 70% | m=1k | 22.84 | 1.26 | 23.78 | 12.34 | 1.26 | 22.45 | 7.90 | 9.00 | 8.83 | 5.03 | 9.37 | 5.09 | 59.53 | 64.12 | 64.69 | 51.36 | 52.32 | 55.39 |
| | | m=2k | 28.48 | 4.64 | 29.64 | 14.92 | 2.55 | 24.69 | 9.08 | 10.91 | 10.00 | 5.84 | 10.37 | 5.96 | 60.57 | 70.46 | 66.26 | 52.86 | 55.75 | 57.26 |
| | | m=3k | 30.49 | 7.02 | 31.35 | 15.37 | 3.37 | 25.00 | 9.69 | 11.77 | 10.50 | 5.84 | 10.55 | 5.92 | 64.61 | 74.48 | 70.66 | 55.76 | 57.80 | 60.05 |
| | | m=4k | 30.38 | 6.91 | 32.76 | 14.06 | 3.53 | 25.58 | 10.08 | 12.13 | 10.98 | 6.00 | 10.69 | 6.03 | 68.09 | 76.77 | 73.13 | 53.57 | 56.62 | 57.78 |
| | | m=5k | 31.90 | 7.67 | 33.73 | 16.93 | 3.98 | 26.53 | 10.53 | 12.24 | 11.35 | 6.06 | 10.54 | 6.34 | 69.58 | 75.94 | 73.29 | 52.69 | 53.91 | 57.57 |
| HPQ | | m=ours | **35.04** | **11.54** | **37.30** | **21.46** | **8.36** | **31.49** | **11.18** | **13.35** | **11.96** | **6.82** | **12.69** | **7.55** | **71.05** | **79.19** | **76.75** | **60.59** | **58.77** | **63.18** |

where ETPQ: Equally Treating Prototype Quantity, AWPQ: Adaptively Weighing Prototype Quantity. Evidently, our AWPQ scheme makes performance improvement.

Table 6: Hybrid-group Prototype Quantity Weighting Effectiveness

| AB | MR | BDGPFEA | | | NUSOBJECT | | | VGGFACEFIFTY | | | VGGFACEHUND | | | YOUTUBEFACE | | | FASHMINST | | |
|---|---|---|---|---|---|---|---|---|---|---|---|---|---|---|---|---|---|---|---|
| | | ACC | NMI | PUR | ACC | NMI | PUR | ACC | NMI | PUR | ACC | NMI | PUR | ACC | NMI | PUR | ACC | NMI | PUR |
| ETHP | 30% | 33.28 | 11.99 | 36.33 | 20.29 | 6.16 | 28.41 | 11.21 | 14.66 | 13.08 | 8.06 | 14.13 | 8.38 | 69.86 | 78.89 | 76.31 | 55.02 | 59.01 | 59.12 |
| AWPQ | | **38.80** | **15.21** | **39.97** | **23.30** | **9.14** | **32.87** | **12.52** | **14.91** | **13.44** | **8.26** | **14.94** | **9.13** | **76.29** | **82.27** | **80.81** | **61.24** | **59.52** | **62.69** |
| ETHP | 50% | 37.70 | 12.37 | 39.04 | 21.37 | 7.24 | 31.79 | 11.29 | 13.44 | 12.13 | 7.11 | 12.87 | 7.54 | 71.32 | 78.93 | 75.67 | 62.08 | 59.90 | **64.84** |
| AWPQ | | **40.31** | **13.88** | **40.31** | **22.38** | **9.21** | **33.92** | **12.31** | **14.48** | **13.21** | **7.55** | **13.85** | **8.39** | **72.60** | **79.60** | **77.65** | **62.51** | **60.22** | 64.64 |
| ETP | 70% | 34.70 | 10.66 | 35.69 | 17.44 | 5.13 | 28.10 | 10.37 | 12.58 | 10.22 | 6.26 | 11.68 | 6.97 | 68.62 | 78.38 | 75.92 | 56.35 | **59.42** | 60.54 |
| AWPQ | | **35.04** | **11.54** | **37.30** | **21.46** | **8.36** | **31.49** | **11.18** | **13.35** | **11.96** | **6.82** | **12.69** | **7.55** | **71.05** | **79.19** | **76.75** | **60.59** | 58.77 | **63.18** |

## 5.5 CONVERGENCE

We draw the objective value evolution of **Algorithm** 2 to illustrate the convergence of SIIHPC. From Fig. 2, we can know that it monotonically decreases and gradually stabilizes within twenty iterations.

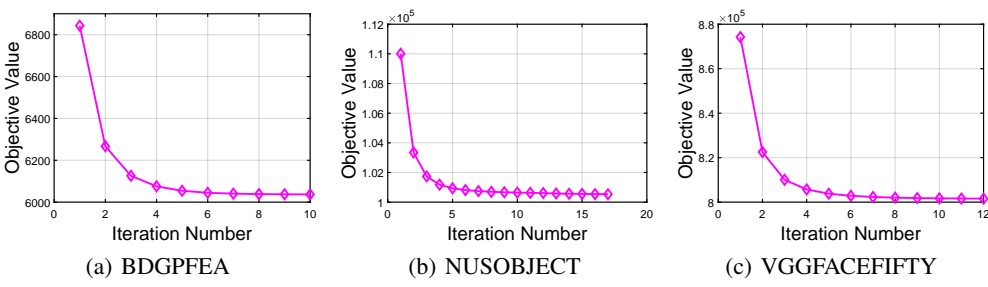

(a) BDGPFEA      (b) NUSOBJECT      (c) VGGFACEFIFTY

Figure 2: The objective value of **Algorithm** 2 on BDGPFEA, NUSOBJECT and VGGFACEFIFTY.

## 5.6 MONOTONICITY OF FUNCTION $g$

To experimentally verify the monotonicity of function $g$, we draw its objective value evolution in Fig. 3. Evidently, $g$ is monotonically increasing. Further, we also give the change of $g$ during each iteration of **Algorithm** 2. Taking Fig. 2 (a) as an example, **Algorithm** 2 iterates totally 10 times. Fig. 4 presents the change of $g$ when **Algorithm** 2 is at the 2-th $\sim$ 10-th iteration respectively. As seen, it is also monotonically increasing. Moreover, it can be observed that as the upper-loop **Algorithm** 2 iterates, the number of iterations required for the inner-loop **Algorithm** 1 gradually decreases, which is mainly because along with the iteration of **Algorithm** 2, the optimization variable in the inner-loop gradually reaches to its optimal solution and accordingly **Algorithm** 1 needs fewer iterations.

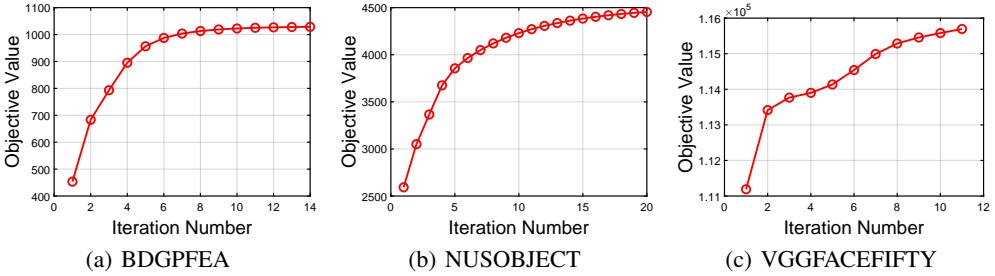

Figure 3: The value change of function $g$ when **Algorithm** 2 is at the 1-th iteration.

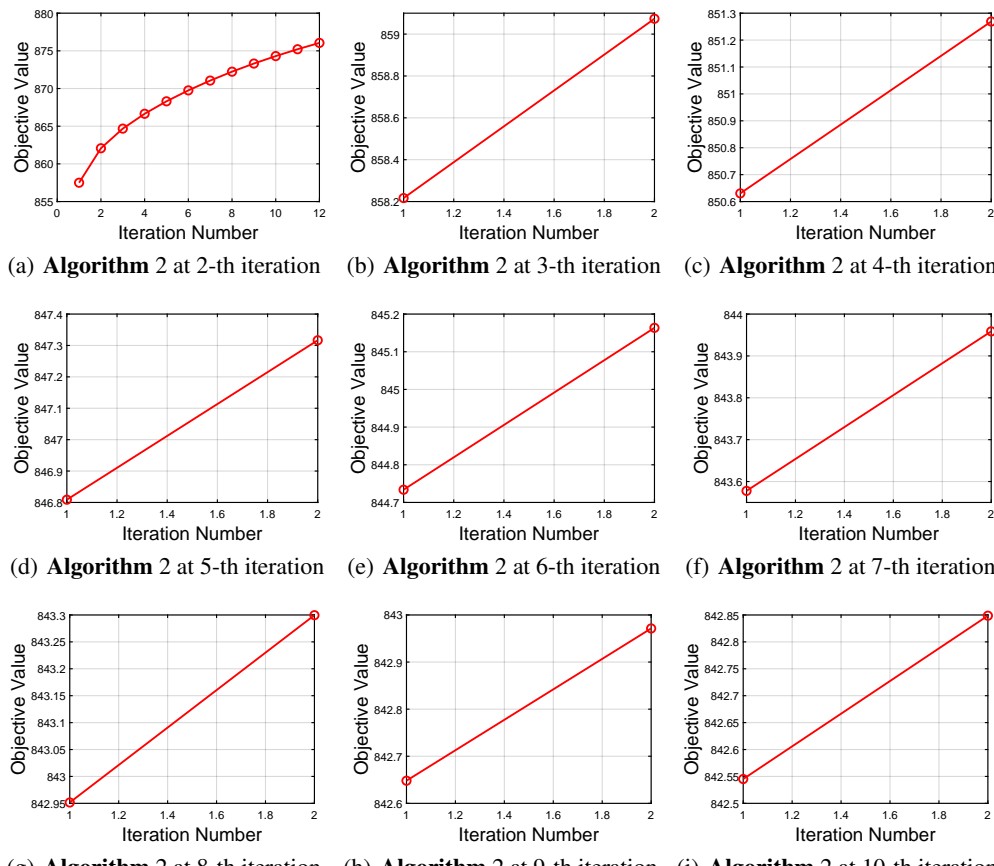

Figure 4: The value change of function $g$ on BDGPFEA.

## 6 CONCLUSION

We propose a simple yet effective IMVC method named SIIHPC in this paper. Rather than only utilizing observed unpaired samples to construct bipartite similarity, it successfully imputes incomplete parts at the similarity level via partial bipartition learning transformation to integrate the latent useful information from missing samples. Then, instead of a single prototype quantity for all views, it introduces a group of hybrid prototype quantities for each view to flexibly extract data features according to the characteristics of respective view itself. To minimize the formulated objective, it carefully decomposes the entire optimization problem into four sub-parts and devises an auxiliary function with theoretically and experimentally proven monotonic-increasing properties. Experimental results on six popular datasets with various missing ratios demonstrate the effectiveness of SIIHPC.

ACKNOWLEDGMENT

This work was supported in part by the Research Grants Council (RGC) Joint Research Scheme under grant: N_HKBU214/21, the General Research Fund of RGC under the grants: 12202622 and 12202924, and the National Natural Science Foundation of China under Grant No. 62406329, 62476280, 62441618, 62325604, 62276271.

ETHICS STATEMENT

This paper does not involve any potential ethics issues.

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

APPENDIX

## A  EXPERIMENTAL SETTING

In the paper, we fine-tune $\lambda$ and $\beta$ in $\{10^{-7}, 10^{-6}, \cdots, 10^{-2}\}$ and $\{10^2, 10^3, \cdots, 10^7\}$ respectively. The hybrid prototype quantities on each view are set as $[1k, 2k, 3k, 4k, 5k]$. To demonstrate the effectiveness of our SIIHPC in clustering incomplete multi-view data, we conduct experiments under missing ratio = 30%, 50% and 70% respectively.

Some reasons about hyper-parameter setting are here. In our model, the hyper-parameters $\lambda$ and $\beta$ aim at fine-tuning $\|\mathbf{G}_s\|_F^2$ and $\|\mathbf{A}\|_F^2$ respectively. Note that $\mathbf{G}_s \in \mathbb{R}^{m_s \times n}$ and $\mathbf{A} \in \mathbb{R}^{V \times S}$ where $m_s$, $n$, $V$ and $S$ denote the numbers of prototypes at the $s$-th group, samples, views and prototype quantity candidates, respectively. Generally, $V \leq m_s$ and $S \ll n$. Therefore, we have that the size of $\mathbf{G}_s$ is much greater than that of $\mathbf{A}$. Besides, combined with the feasible regions, we have that the absolute values of the elements in $\mathbf{G}_s$ and $\mathbf{A}$ are all within the range $[0, 1]$, Therefore, we can get that $\|\mathbf{G}_s\|_F^2$ is much larger than $\|\mathbf{A}\|_F^2$. Based on these, we search $\lambda$ within a small range while searching $\beta$ within a relatively large range.

## B  SYMBOL SUMMARY

$V$: the quantity of views;

$S$: the element quantity of hybrid prototype group;

$d_v$: the feature dimension on view $v$;

$n$: the quantity of samples;

$k$: the quantity of clusters;

$n_v$: the quantity of observed samples on view $v$;

$m_s$: the $s$-th prototype quantity on each view;

$\mathbf{D}_v$: the data matrix on view $v$, $d_v \times n$;

$\mathbf{X}_v$: the incomplete similarity matrix on view $v$, $m \times n$;

$\mathbf{X}_{v,s}$: the incomplete similarity with the $s$-th scale on view $v$, $m_s \times n$;

$\mathbf{W}_v$: the constructed index matrix on view $v$, $n \times n_v$;

$\mathbf{H}_v$: the prototype matrix on view $v$, $d_v \times m$;

$\mathbf{H}_{v,s}$: the prototype matrix with the $s$-th quantity on view $v$, $d_v \times m_s$;

$\mathbf{Q}_{v,s}$: the imputation matrix with the $s$-th scale on view $v$, $m_s \times n$;

$\mathbf{M}_v$: the constructed index matrix on view $v$, $n \times (n - n_v)$;

$\mathbf{G}$: the consensus graph, $m \times n$;

$\mathbf{G}_s$: the consensus graph with the $s$-th scale, $m_s \times n$;

$\mathbf{A}$: the prototype quantity weighting matrix, $V \times S$;

$\widehat{\mathbf{L}}_v$: temporary variable, $d_v \times d_v$;

$\mathbf{L}_v$: temporary variable, $d_v \times d_v$;

$\mathbf{P}_{v,s}$: temporary variable, $d_v \times m_s$;

$\mathbf{J}_{v,s}$: temporary variable, $m_s \times n$;

$\mathbf{F}_{v,s}$: temporary variable, $m_s \times n$.

## C    Terminology Summary

In the paper, we utilize some terms like T-PBL, IVHGP, OS, MSVSG, etc. Here, we provide more detailed descriptions about them.

T-PBL represents transforming partial bipartition learning into original sample form to split out of observed similarity.

IVHGP represents introducing a group of hybrid prototype quantities for each individual view to flexibly extract the data features belonging to each view itself, i.e., intra-view hybrid-group prototypes.

OS represents the observed similarity on each individual view.

MSVSG represents the generated graphs with various scales on each view, i.e., multi-scale view-specific graphs.

SLI represents learning to recover the incomplete parts by utilizing the connection built between the similarity exclusive on respective view and the consensus graph shared for all views, i.e., similarity-level imputation.

CG represents the learned consensus graph that is shared for all views.

SG represents conducting spectral grouping on the graph generated by stacking multi-scale consensus graphs.

## D    Proof of Lemma 1

Recall the function $g$,

$$g(\mathbf{H}_{v,s}) = \mathrm{Tr}\left(\mathbf{H}_{v,s}^\top \widehat{\mathbf{L}}_v \mathbf{H}_{v,s} + \mathbf{H}_{v,s}^\top \mathbf{P}_{v,s}\right), \tag{12}$$

where $\widehat{\mathbf{L}}_v = \varphi_v \mathbf{I}_{d_v} - \mathbf{L}_v$, $\mathbf{L}_v = \mathbf{D}_v \mathbf{W}_v \mathbf{W}_v^\top \mathbf{W}_v \mathbf{W}_v^\top \mathbf{D}_v^\top$, and then, together with the symmetry of $\widehat{\mathbf{L}}_v$, we have that its derivative $\nabla g(\mathbf{H}_{v,s})$ is

$$\nabla g(\mathbf{H}_{v,s}) = \widehat{\mathbf{L}}_v \mathbf{H}_{v,s} + \widehat{\mathbf{L}}_v^\top \mathbf{H}_{v,s} + \mathbf{P}_{v,s} = 2\widehat{\mathbf{L}}_v \mathbf{H}_{v,s} + \mathbf{P}_{v,s}. \tag{13}$$

Denote the value of $\mathbf{H}_{v,s}$ at the $r$-th iteration as $(\mathbf{H}_{v,s})^r$, the SVD of $\nabla g((\mathbf{H}_{v,s})^r)$ as $(\mathbf{U}_{v,s})^r (\mathbf{\Sigma}_{v,s})^r (\mathbf{V}_{v,s}^\top)^r$. Then, for the $\mathbf{H}_{v,s}$ at any iteration number, we have the following equality holds,

$$\begin{aligned}
\mathrm{Tr}\left(\mathbf{H}_{v,s}^\top \nabla g((\mathbf{H}_{v,s})^r)\right) &= \mathrm{Tr}\left(\mathbf{H}_{v,s}^\top (\mathbf{U}_{v,s})^r (\mathbf{\Sigma}_{v,s})^r (\mathbf{V}_{v,s}^\top)^r\right) \\
&= \mathrm{Tr}\left((\mathbf{V}_{v,s}^\top)^r \mathbf{H}_{v,s}^\top (\mathbf{U}_{v,s})^r (\mathbf{\Sigma}_{v,s})^r\right).
\end{aligned} \tag{14}$$

Considering that $(\mathbf{V}_{v,s}^\top)^r \mathbf{H}_{v,s}^\top (\mathbf{U}_{v,s})^r (\mathbf{U}_{v,s}^\top)^r \mathbf{H}_{v,s} (\mathbf{V}_{v,s})^r = \mathbf{I}$, we have $(\mathbf{V}_{v,s}^\top)^r \mathbf{H}_{v,s}^\top (\mathbf{U}_{v,s})^r$ is orthogonal. Additionally, the diagonal elements of $(\mathbf{\Sigma}_{v,s})^r$ are all non-negative. Therefore, we have

$$\mathrm{Tr}\left(\mathbf{H}_{v,s}^\top \nabla g((\mathbf{H}_{v,s})^r)\right) \leq \mathrm{Tr}\left((\mathbf{\Sigma}_{v,s})^r\right). \tag{15}$$

In particular, when $(\mathbf{V}_{v,s}^\top)^r \mathbf{H}_{v,s}^\top (\mathbf{U}_{v,s})^r = \mathbf{I}$, i.e., $\mathbf{H}_{v,s}$ takes $(\mathbf{U}_{v,s})^r (\mathbf{V}_{v,s}^\top)^r$, the equality holds. Therefore, under $(\mathbf{H}_{v,s})^{r+1}$ taking $(\mathbf{U}_{v,s})^r (\mathbf{V}_{v,s}^\top)^r$, we have the following inequality always holds,

$$\mathrm{Tr}\left((\mathbf{H}_{v,s}^\top)^{r+1} \nabla g((\mathbf{H}_{v,s})^r)\right) \geq \mathrm{Tr}\left(\mathbf{H}_{v,s}^\top \nabla g((\mathbf{H}_{v,s})^r)\right). \tag{16}$$

At the $(r+1)$-th iteration, accordingly, we also always have

$$\mathrm{Tr}\left((\mathbf{H}_{v,s}^\top)^{r+1} \nabla g((\mathbf{H}_{v,s})^r)\right) - \mathrm{Tr}\left((\mathbf{H}_{v,s}^\top)^r \nabla g((\mathbf{H}_{v,s})^r)\right) \geq 0. \tag{17}$$

Consequently, **Lemma** 1 holds.

# E    PROOF OF LEMMA 2

According to the fact that $\widehat{\mathbf{L}}_v$ is positive semi-definite, we have $\widehat{\mathbf{L}}_v$ can be expressed as $\mathbf{Y}\,\mathbf{Y}^\top$ via matrix decomposition. On this basis, we can further obtain

$$
\begin{aligned}
&\mathrm{Tr}\left(\left[(\mathbf{H}_{v,s})^{r+1}-(\mathbf{H}_{v,s})^{r}\right]^\top \widehat{\mathbf{L}}_v \left[(\mathbf{H}_{v,s})^{r+1}-(\mathbf{H}_{v,s})^{r}\right]\right) = \\
&\mathrm{Tr}\left(\left[(\mathbf{H}_{v,s})^{r+1}-(\mathbf{H}_{v,s})^{r}\right]^\top \mathbf{Y}\mathbf{Y}^\top \left[(\mathbf{H}_{v,s})^{r+1}-(\mathbf{H}_{v,s})^{r}\right]\right) = \\
&\left\|\left[(\mathbf{H}_{v,s})^{r+1}-(\mathbf{H}_{v,s})^{r}\right]^\top \mathbf{Y}\right\|_F^2 \geq 0.
\end{aligned}
\tag{18}
$$

Therefore, we have the following inequality holds,

$$
\begin{aligned}
&\mathrm{Tr}\left((\mathbf{H}_{v,s}^\top)^{r+1}\widehat{\mathbf{L}}_v (\mathbf{H}_{v,s})^{r+1} - (\mathbf{H}_{v,s}^\top)^{r}\widehat{\mathbf{L}}_v (\mathbf{H}_{v,s})^{r+1}\right.\\
&\left.\quad - (\mathbf{H}_{v,s}^\top)^{r+1}\widehat{\mathbf{L}}_v (\mathbf{H}_{v,s})^{r} + (\mathbf{H}_{v,s}^\top)^{r}\widehat{\mathbf{L}}_v (\mathbf{H}_{v,s})^{r}\right) = \\
&\mathrm{Tr}\left(\left[(\mathbf{H}_{v,s}^\top)^{r+1}-(\mathbf{H}_{v,s}^\top)^{r}\right]\widehat{\mathbf{L}}_v \left[(\mathbf{H}_{v,s})^{r+1}-(\mathbf{H}_{v,s})^{r}\right]\right) = \\
&\mathrm{Tr}\left(\left[(\mathbf{H}_{v,s})^{r+1}-(\mathbf{H}_{v,s})^{r}\right]^\top \widehat{\mathbf{L}}_v \left[(\mathbf{H}_{v,s})^{r+1}-(\mathbf{H}_{v,s})^{r}\right]\right) \geq 0.
\end{aligned}
\tag{19}
$$

Merging common items yields

$$
\mathrm{Tr}\left((\mathbf{H}_{v,s}^\top)^{r+1}\widehat{\mathbf{L}}_v \left[(\mathbf{H}_{v,s})^{r+1}-(\mathbf{H}_{v,s})^{r}\right] - (\mathbf{H}_{v,s}^\top)^{r}\widehat{\mathbf{L}}_v \left[(\mathbf{H}_{v,s})^{r+1}-(\mathbf{H}_{v,s})^{r}\right]\right) \geq 0. \tag{20}
$$

Furthermore, in conjunction with the symmetry of $\widehat{\mathbf{L}}_v$, we have

$$
\mathrm{Tr}\left((\mathbf{H}_{v,s}^\top)^{r}\widehat{\mathbf{L}}_v \left[(\mathbf{H}_{v,s})^{r+1}-(\mathbf{H}_{v,s})^{r}\right]\right) = \mathrm{Tr}\left(\left[(\mathbf{H}_{v,s}^\top)^{r+1}-(\mathbf{H}_{v,s}^\top)^{r}\right]\widehat{\mathbf{L}}_v (\mathbf{H}_{v,s})^{r}\right). \tag{21}
$$

Therefore, we have the following inequality holds,

$$
\mathrm{Tr}\left((\mathbf{H}_{v,s}^\top)^{r+1}\widehat{\mathbf{L}}_v \left[(\mathbf{H}_{v,s})^{r+1}-(\mathbf{H}_{v,s})^{r}\right] - \left[(\mathbf{H}_{v,s}^\top)^{r+1}-(\mathbf{H}_{v,s}^\top)^{r}\right]\widehat{\mathbf{L}}_v (\mathbf{H}_{v,s})^{r}\right) \geq 0. \tag{22}
$$

Consequently, **Lemma** 2 holds.

# F    PROOF OF THEOREM 1

Proving that the function $g(\mathbf{H}_{v,s})$ is monotonically increasing is equivalent to proving $g\left((\mathbf{H}_{v,s})^{r+1}\right) \geq g\left((\mathbf{H}_{v,s})^{r}\right)$ for any iteration number $r$.

According to **Lemma** 1 and the derivative of function $g$, we can get

$$
\mathrm{Tr}\left(\left[(\mathbf{H}_{v,s}^\top)^{r+1}-(\mathbf{H}_{v,s}^\top)^{r}\right]\left(2\widehat{\mathbf{L}}_v (\mathbf{H}_{v,s})^{r} + \mathbf{P}_{v,s}\right)\right) \geq 0. \tag{23}
$$

Therefore, the following inequality holds,

$$
\begin{aligned}
&\mathrm{Tr}\left((\mathbf{H}_{v,s}^\top)^{r+1}\left(2\widehat{\mathbf{L}}_v (\mathbf{H}_{v,s})^{r} + \mathbf{P}_{v,s}\right) - (\mathbf{H}_{v,s}^\top)^{r}\widehat{\mathbf{L}}_v (\mathbf{H}_{v,s})^{r}\right) \geq \\
&\mathrm{Tr}\left((\mathbf{H}_{v,s}^\top)^{r}\widehat{\mathbf{L}}_v (\mathbf{H}_{v,s})^{r} + (\mathbf{H}_{v,s}^\top)^{r}\mathbf{P}_{v,s}\right).
\end{aligned}
\tag{24}
$$

According to **Lemma** 2, we can get

$$
\begin{aligned}
&\mathrm{Tr}\left((\mathbf{H}_{v,s}^\top)^{r+1}\widehat{\mathbf{L}}_v \left[(\mathbf{H}_{v,s})^{r+1}-(\mathbf{H}_{v,s})^{r}\right] + (\mathbf{H}_{v,s}^\top)^{r+1}\mathbf{P}_{v,s}\right) \geq \\
&\mathrm{Tr}\left(\left[(\mathbf{H}_{v,s}^\top)^{r+1}-(\mathbf{H}_{v,s}^\top)^{r}\right]\widehat{\mathbf{L}}_v (\mathbf{H}_{v,s})^{r} + (\mathbf{H}_{v,s}^\top)^{r+1}\mathbf{P}_{v,s}\right).
\end{aligned}
\tag{25}
$$

Therefore, we can have

$$
\begin{aligned}
&\mathrm{Tr}\left(\left(\mathbf{H}_{v,s}^{\top}\right)^{r+1}\widehat{\mathbf{L}}_{v}\left(\mathbf{H}_{v,s}\right)^{r+1}+\left(\mathbf{H}_{v,s}^{\top}\right)^{r+1}\mathbf{P}_{v,s}\right)\geq \\
&\mathrm{Tr}\left(2\left(\mathbf{H}_{v,s}^{\top}\right)^{r+1}\widehat{\mathbf{L}}_{v}\left(\mathbf{H}_{v,s}\right)^{r}-\left(\mathbf{H}_{v,s}^{\top}\right)^{r}\widehat{\mathbf{L}}_{v}\left(\mathbf{H}_{v,s}\right)^{r}+\left(\mathbf{H}_{v,s}^{\top}\right)^{r+1}\mathbf{P}_{v,s}\right).
\end{aligned}
\tag{26}
$$

Further, we have the following equality always holds,

$$
\begin{aligned}
&\mathrm{Tr}\left(2\left(\mathbf{H}_{v,s}^{\top}\right)^{r+1}\widehat{\mathbf{L}}_{v}\left(\mathbf{H}_{v,s}\right)^{r}-\left(\mathbf{H}_{v,s}^{\top}\right)^{r}\widehat{\mathbf{L}}_{v}\left(\mathbf{H}_{v,s}\right)^{r}+\left(\mathbf{H}_{v,s}^{\top}\right)^{r+1}\mathbf{P}_{v,s}\right)= \\
&\mathrm{Tr}\left(\left(\mathbf{H}_{v,s}^{\top}\right)^{r+1}\left(2\widehat{\mathbf{L}}_{v}\left(\mathbf{H}_{v,s}\right)^{r}+\mathbf{P}_{v,s}\right)-\left(\mathbf{H}_{v,s}^{\top}\right)^{r}\widehat{\mathbf{L}}_{v}\left(\mathbf{H}_{v,s}\right)^{r}\right).
\end{aligned}
\tag{27}
$$

Noticed that $g((\mathbf{H}_{v,s})^{r}) = \mathrm{Tr}\left(\left(\mathbf{H}_{v,s}^{\top}\right)^{\top}\widehat{\mathbf{L}}_{v}\left(\mathbf{H}_{v,s}\right)^{r}+\left(\mathbf{H}_{v,s}^{\top}\right)^{r}\mathbf{P}_{v,s}\right)$ and $g((\mathbf{H}_{v,s})^{r+1}) = \mathrm{Tr}\left(\left(\mathbf{H}_{v,s}^{\top}\right)^{r+1}\widehat{\mathbf{L}}_{v}\left(\mathbf{H}_{v,s}\right)^{r+1}+\left(\mathbf{H}_{v,s}^{\top}\right)^{r+1}\mathbf{P}_{v,s}\right)$, and in conjunction with (24), (26) and (27), we can get

$$
\mathrm{Tr}\left(\left(\mathbf{H}_{v,s}^{\top}\right)^{r+1}\widehat{\mathbf{L}}_{v}\left(\mathbf{H}_{v,s}\right)^{r+1}+\left(\mathbf{H}_{v,s}^{\top}\right)^{r+1}\mathbf{P}_{v,s}\right)\geq\mathrm{Tr}\left(\left(\mathbf{H}_{v,s}^{\top}\right)^{r}\widehat{\mathbf{L}}_{v}\left(\mathbf{H}_{v,s}\right)^{r}+\left(\mathbf{H}_{v,s}^{\top}\right)^{r}\mathbf{P}_{v,s}\right).
\tag{28}
$$

That is, $g\left(\left(\mathbf{H}_{v,s}\right)^{r+1}\right)\geq g\left(\left(\mathbf{H}_{v,s}\right)^{r}\right)$. The proof is complete.

# G    DERIVATION PROCEDURE OF OPTIMIZATION VARIABLES

## G.1    *Optimizing the Prototype Matrix $\boldsymbol{H}_{v,s}$*

Fixing $\mathbf{Q}_{v,s}$, $\mathbf{G}_{s}$ and $\mathbf{A}$, we can equivalently transform the original optimization problem (2) as

$$
\begin{aligned}
&\min_{\mathbf{H}_{v,s}}\left\|\mathbf{H}_{v,s}^{\top}\mathbf{D}_{v}\mathbf{W}_{v}\mathbf{W}_{v}^{\top}+\mathbf{Q}_{v,s}\mathbf{M}_{v}\mathbf{M}_{v}^{\top}-\mathbf{G}_{s}\right\|_{F}^{2} \\
&s.t.\ \ \mathbf{H}_{v,s}^{\top}\mathbf{H}_{v,s}=\mathbf{I}_{m_{s}}.
\end{aligned}
\tag{29}
$$

Using trace operation, we can obtain

$$
\begin{aligned}
&\min_{\mathbf{H}_{v,s}}\left\|\mathbf{H}_{v,s}^{\top}\mathbf{D}_{v}\mathbf{W}_{v}\mathbf{W}_{v}^{\top}+\mathbf{Q}_{v,s}\mathbf{M}_{v}\mathbf{M}_{v}^{\top}-\mathbf{G}_{s}\right\|_{F}^{2}\Leftrightarrow \\
&\min_{\mathbf{H}_{v,s}}\mathrm{Tr}\left(\mathbf{H}_{v,s}^{\top}\mathbf{L}_{v}\mathbf{H}_{v,s}+2\mathbf{H}_{v,s}^{\top}\mathbf{D}_{v}\mathbf{W}_{v}\mathbf{W}_{v}^{\top}\left(\mathbf{Q}_{v,s}\mathbf{M}_{v}\mathbf{M}_{v}^{\top}-\mathbf{G}_{s}\right)^{\top}\right),
\end{aligned}
\tag{30}
$$

where $\mathbf{L}_{v}=\mathbf{D}_{v}\mathbf{W}_{v}\mathbf{W}_{v}^{\top}\mathbf{W}_{v}\mathbf{W}_{v}^{\top}\mathbf{D}_{v}^{\top}$. Denote the largest eigenvalue of $\mathbf{L}_{v}$ as $\varphi_{v}$, and in conjunction with $\mathbf{H}_{v,s}^{\top}\mathbf{H}_{v,s}=\mathbf{I}_{m_{s}}$, we have

$$
\begin{aligned}
&\min_{\mathbf{H}_{v,s}}\mathrm{Tr}\left(\mathbf{H}_{v,s}^{\top}\mathbf{L}_{v}\mathbf{H}_{v,s}+2\mathbf{H}_{v,s}^{\top}\mathbf{D}_{v}\mathbf{W}_{v}\mathbf{W}_{v}^{\top}\left(\mathbf{Q}_{v,s}\mathbf{M}_{v}\mathbf{M}_{v}^{\top}-\mathbf{G}_{s}\right)^{\top}\right)\Leftrightarrow \\
&\min_{\mathbf{H}_{v,s}}\mathrm{Tr}\left(\mathbf{H}_{v,s}^{\top}\left(\mathbf{L}_{v}-\varphi_{v}\mathbf{I}_{d_{v}}\right)\mathbf{H}_{v,s}+2\mathbf{H}_{v,s}^{\top}\mathbf{D}_{v}\mathbf{W}_{v}\mathbf{W}_{v}^{\top}\left(\mathbf{Q}_{v,s}\mathbf{M}_{v}\mathbf{M}_{v}^{\top}-\mathbf{G}_{s}\right)^{\top}\right)\Leftrightarrow \\
&\max_{\mathbf{H}_{v,s}}\mathrm{Tr}\left(\mathbf{H}_{v,s}^{\top}\widehat{\mathbf{L}}_{v}\mathbf{H}_{v,s}+\mathbf{H}_{v,s}^{\top}\mathbf{P}_{v,s}\right),
\end{aligned}
\tag{31}
$$

where $\widehat{\mathbf{L}}_{v}=\varphi_{v}\mathbf{I}_{d_{v}}-\mathbf{L}_{v}$, $\mathbf{P}_{v,s}=2\mathbf{D}_{v}\mathbf{W}_{v}\mathbf{W}_{v}^{\top}\left(\mathbf{G}_{s}-\mathbf{Q}_{v,s}\mathbf{M}_{v}\mathbf{M}_{v}^{\top}\right)^{\top}$. Then, in conjunction with **Theorem** 1, we can obtain that the objective value in (31) is monotonically increasing. Therefore, we can determine the value of $\mathbf{H}_{v,s}$ by comparing the objective values at $\left(\mathbf{H}_{v,s}\right)^{r+1}$ and $\left(\mathbf{H}_{v,s}\right)^{r}$ where $r$ denotes the $r$-th iteration.

## G.2    *Optimizing the Similarity Imputation Matrix $\boldsymbol{Q}_{v,s}$*

Fixing $\mathbf{H}_{v,s}$, $\mathbf{G}_{s}$ and $\mathbf{A}$, we can equivalently transform the optimization problem (2) as

$$
\begin{aligned}
&\min_{\mathbf{Q}_{v,s}}\left\|\mathbf{H}_{v,s}^{\top}\mathbf{D}_{v}\mathbf{W}_{v}\mathbf{W}_{v}^{\top}+\mathbf{Q}_{v,s}\mathbf{M}_{v}\mathbf{M}_{v}^{\top}-\mathbf{G}_{s}\right\|_{F}^{2} \\
&s.t.\ \ -1\leq\mathbf{Q}_{v,s}\leq1.
\end{aligned}
\tag{32}
$$

Denote the matrix $\mathbf{J}_{v,s} = \mathbf{G}_s - \mathbf{H}_{v,s}^\top \mathbf{D}_v \mathbf{W}_v \mathbf{W}_v^\top$, and then we can obtain that the value $\left\| \mathbf{Q}_{v,s} \mathbf{M}_v \mathbf{M}_v^\top - \mathbf{J}_{v,s} \right\|_F^2$ is equal to $\left\| \mathbf{H}_{v,s}^\top \mathbf{D}_v \mathbf{W}_v \mathbf{W}_v^\top + \mathbf{Q}_{v,s} \mathbf{M}_v \mathbf{M}_v^\top - \mathbf{G}_s \right\|_F^2$. Accordingly, the objective value of (32) can be minimizing by making $\mathbf{Q}_{v,s} \mathbf{M}_v \mathbf{M}_v^\top$ and $\mathbf{J}_{v,s}$ as close as possible. Therefore, we can determine $\mathbf{Q}_{v,s}$ using the value of corresponding index of $\mathbf{J}_{v,s}$. Subsequently, to ensure that the solution is within the feasible region, we regularize it by first comparing it and $\pm 1$ and then conducting truncation.

### G.3 *Optimizing the Unified Representation Matrix $\mathbf{G}_s$*

Fixing $\mathbf{H}_{v,s}$, $\mathbf{Q}_{v,s}$ and $\mathbf{A}$, we can equivalently transform the problem (2) as

$$\min_{\mathbf{G}_s} \sum_{v=1}^{V} a_{v,s} \left( \left\| \mathbf{H}_{v,s}^\top \mathbf{D}_v \mathbf{W}_v \mathbf{W}_v^\top + \mathbf{Q}_{v,s} \mathbf{M}_v \mathbf{M}_v^\top - \mathbf{G}_s \right\|_F^2 + \lambda \left\| \mathbf{G}_s \right\|_F^2 \right)$$

$$s.t. \ -1 \leq \mathbf{G}_s \leq 1. \tag{33}$$

Expanding the objective using trace operation yields

$$\min_{\mathbf{G}_s} \sum_{v=1}^{V} a_{v,s} \left( \left\| \mathbf{H}_{v,s}^\top \mathbf{D}_v \mathbf{W}_v \mathbf{W}_v^\top + \mathbf{Q}_{v,s} \mathbf{M}_v \mathbf{M}_v^\top - \mathbf{G}_s \right\|_F^2 + \lambda \left\| \mathbf{G}_s \right\|_F^2 \right) \Leftrightarrow$$

$$\min_{\mathbf{G}_s} \sum_{v=1}^{V} a_{v,s} \left( \mathrm{Tr} \left( \mathbf{G}_s^\top (1 + \lambda) \mathbf{I}_{m_s} \mathbf{G}_s - 2 \mathbf{F}_{v,s}^\top \mathbf{G}_s \right) \right) \Leftrightarrow \tag{34}$$

$$\min_{\mathbf{G}_s} \mathrm{Tr} \left( \mathbf{G}_s^\top \sum_{v=1}^{V} a_{v,s} (1 + \lambda) \mathbf{I}_{m_s} \mathbf{G}_s - 2 \sum_{v=1}^{V} \left( a_{v,s} \mathbf{F}_{v,s} \right)^\top \mathbf{G}_s \right),$$

where the matrix $\mathbf{F}_{v,s} = \mathbf{H}_{v,s}^\top \mathbf{D}_v \mathbf{W}_v \mathbf{W}_v^\top + \mathbf{Q}_{v,s} \mathbf{M}_v \mathbf{M}_v^\top$. After expanding the trace by elements, we can obtain

$$\min_{\mathbf{G}_s} \mathrm{Tr} \left( \mathbf{G}_s^\top \sum_{v=1}^{V} a_{v,s} (1 + \lambda) \mathbf{I}_{m_s} \mathbf{G}_s - 2 \sum_{v=1}^{V} \left( a_{v,s} \mathbf{F}_{v,s} \right)^\top \mathbf{G}_s \right) \Leftrightarrow$$

$$\min_{[\mathbf{G}_s]_{:,j}} [\mathbf{G}_s]_{:,j}^\top \sum_{v=1}^{V} a_{v,s} (1 + \lambda) \mathbf{I}_{m_s} [\mathbf{G}_s]_{:,j} - 2 \left( \sum_{v=1}^{V} a_{v,s} [\mathbf{F}_{v,s}]_{:,j} \right)^\top [\mathbf{G}_s]_{:,j}. \tag{35}$$

Consequently, together with (34) and (35), we can further transform (33) as

$$\min_{[\mathbf{G}_s]_{:,j}} [\mathbf{G}_s]_{:,j}^\top \sum_{v=1}^{V} a_{v,s} (1 + \lambda) \mathbf{I}_{m_s} [\mathbf{G}_s]_{:,j} - 2 \left( \sum_{v=1}^{V} a_{v,s} [\mathbf{F}_{v,s}]_{:,j} \right)^\top [\mathbf{G}_s]_{:,j}$$

$$s.t. \ -1 \leq [\mathbf{G}_s]_{:,j} \leq 1, \tag{36}$$

which is a classical quadratic programming problem, and can be solved by present software packages.

### G.4 *Optimizing the Prototype Balance Matrix $\mathbf{A}$*

Fixing $\mathbf{H}_{v,s}$, $\mathbf{Q}_{v,s}$ and $\mathbf{G}_s$, we can transform the problem (2) as

$$\min_{\mathbf{A}} \sum_{v=1}^{V} \sum_{s=1}^{S} a_{v,s} \left( \left\| \mathbf{H}_{v,s}^\top \mathbf{D}_v \mathbf{W}_v \mathbf{W}_v^\top + \mathbf{Q}_{v,s} \mathbf{M}_v \mathbf{M}_v^\top - \mathbf{G}_s \right\|_F^2 + \lambda \left\| \mathbf{G}_s \right\|_F^2 \right) + \beta \left\| \mathbf{A} \right\|_F^2$$

$$s.t. \ \mathbf{A1} = \mathbf{1}, 0 \leq \mathbf{A}. \tag{37}$$

The scalar $p_{v,s} = \left\| \mathbf{H}_{v,s}^\top \mathbf{D}_v \mathbf{W}_v \mathbf{W}_v^\top + \mathbf{Q}_{v,s} \mathbf{M}_v \mathbf{M}_v^\top - \mathbf{G}_s \right\|_F^2 + \lambda \left\| \mathbf{G}_s \right\|_F^2$ is a constant with respect to $a_{v,s}$. Besides, the constraints are for the rows of $\mathbf{A}$, and therefore we optimize $\mathbf{A}$ row by row. In

particular, we can equivalently transform (37) as

$$\min_{\mathbf{a}_{v,:}} \sum_{s=1}^{S} a_{v,s} p_{v,s} + \beta \left\| \mathbf{a}_{v,:} \right\|_F^2 \tag{38}$$
$$s.t. \ \mathbf{a}_{v,:} \mathbf{1} = 1, 0 \le \mathbf{a}_{v,:},$$

where the vector $\mathbf{a}_{v,:}$ denotes the $v$-th row of $\mathbf{A}$. We can further transform (38) as

$$\min_{\mathbf{a}_{v,:}} \left\| \mathbf{a}_{v,:} + \frac{\mathbf{p}_{v,:}}{2\beta} \right\|_F^2 \tag{39}$$
$$s.t. \ \mathbf{a}_{v,:} \mathbf{1} = 1, 0 \le \mathbf{a}_{v,:},$$

where the vector $\mathbf{p}_{v,:}$ consists of $p_{v,s}$, $s = 1, 2, \cdots, S$. The Lagrangian function of (39) is

$$\mathcal{L}\left(\mathbf{a}_{v,:}, \kappa, \boldsymbol{\mu}\right) = \frac{1}{2} \left\| \mathbf{a}_{v,:} + \frac{\mathbf{p}_{v,:}}{2\beta} \right\|_F^2 + \kappa \left(\mathbf{a}_{v,:} \mathbf{1} - 1\right) + \mathbf{a}_{v,:} \boldsymbol{\mu}, \tag{40}$$

where the scalar $\kappa$ and vector $\boldsymbol{\mu} \in \mathbb{R}^S$ denote Lagrangian Multipliers. In conjunction with KKT conditions, we can obtain

$$\mathbf{a}_{v,:} + \frac{\mathbf{p}_{v,:}}{2\beta} + \kappa \mathbf{1}^\top + \boldsymbol{\mu}^\top = 0, \ \ \mathbf{a}_{v,:} \odot \boldsymbol{\mu}^\top = \mathbf{0}^\top. \tag{41}$$

Therefore, we have

$$\mathbf{a}_{v,:} = -\frac{\mathbf{p}_{v,:}}{2\beta} - \kappa \mathbf{1}^\top - \boldsymbol{\mu}^\top, \ \ a_{v,s}\mu_s = 0, s = 1, 2, \cdots, S. \tag{42}$$

Then, in conjunction with the constraint of row sum, we can obtain

$$\mathbf{a}_{v,:} = \left( -\frac{\mathbf{p}_{v,:}}{2\beta} - \kappa \mathbf{1}^\top \right)_+, \ \ \kappa = \frac{-1 - \frac{\mathbf{p}_{v,:}\mathbf{1}}{2\beta}}{S}. \tag{43}$$

Therefore, we have $\mathbf{a}_{v,:} = \left( \frac{\frac{\mathbf{p}_{v,:}\cdot\mathbf{1}\cdot\mathbf{1}^\top}{2\beta} + \mathbf{1}^\top}{S} - \frac{\mathbf{p}_{v,:}}{2\beta} \right)_+.$

## H  OTHER CONVERGENCE EXAMPLES

During alternatively updating the optimization variables $\mathbf{H}_{v,s}$, $\mathbf{Q}_{v,s}$, $\mathbf{G}_s$ and $\mathbf{A}$, their solutions can be acquired by Algorithm 1, (6), (7) and (11) respectively. This indicates that the objective value of Algorithm 2 is monotonically reducing during optimizing each variable. Besides, the objective in (2) has the lower bound, for example 0. In conjunction with the monotonic-reducing and lower bound properties, therefore, our **Algorithm** 2 is convergent.

Besides the convergence examples in Fig. 2, we also present the change in the objective value of **Algorithm** 2 on datasets VGGFACEHUND, YOUBUBEFACE and FASHMNIST, as shown in Fig. 5. It can be seen that these examples are also convergent.

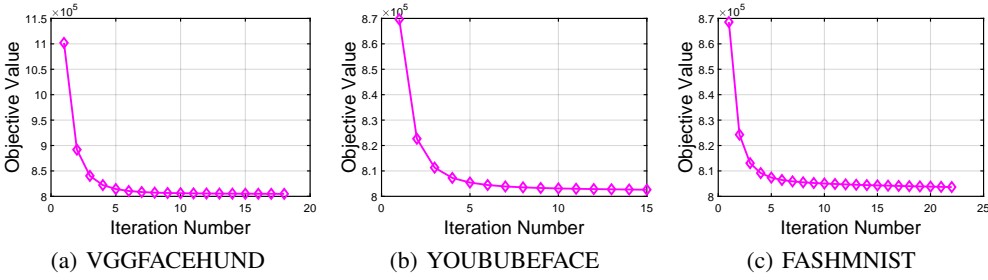

(a) VGGFACEHUND  (b) YOUBUBEFACE  (c) FASHMNIST

Figure 5: The objective value of **Algorithm** 2 on VGGFACEHUND, YOUBUBEFACE and FASHM-NIST.

# I   MORE EXAMPLES ABOUT THE FUNCTION $g$

The function $g$ is monotonically increasing. To further illustrate this point, besides the examples in Fig. 3 and 4 we also give its objective value evolution on datasets VGGFACEHUND, YOUBUBE-FACE and FASHMNIST, as shown in Fig. 6 and Fig 10. It is easy to see that the function $g$ is indeed monotonically increasing.

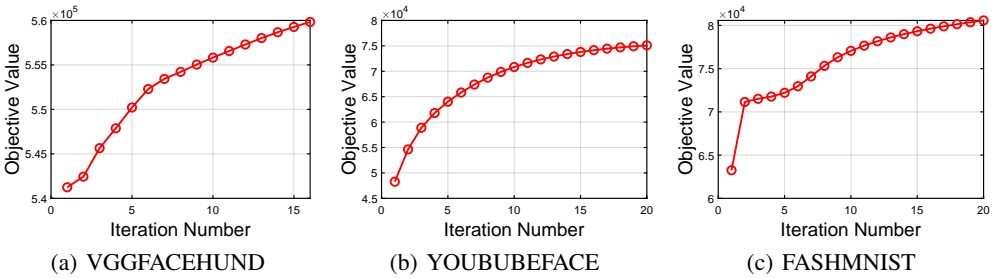

| (a) VGGFACEHUND | (b) YOUBUBEFACE | (c) FASHMNIST |

Figure 6:  The value change of function $g$ on VGGFACEHUND, YOUBUBEFACE and FASHMNIST when **Algorithm** 2 is at the 1-th iteration.

Table 7: Performance Comparison under Missing Ratio = 40%

| Method | BDGPFEA | | | NUSOBJECT | | | VGGFACEFIFTY | | | VGGFACEHUND | | | YOUTUBEFACE | | | FASHMINST | | |
|---|---|---|---|---|---|---|---|---|---|---|---|---|---|---|---|---|---|---|
| | ACC | NMI | PUR | ACC | NMI | PUR | ACC | NMI | PUR | ACC | NMI | PUR | ACC | NMI | PUR | ACC | NMI | PUR |
| LSIMVC | 26.68 | 6.80 | 26.68 | 21.65 | 8.92 | 31.54 | 7.55 | 10.11 | 7.78 | | | | | | | | | |
| GSRIMC | 39.03 | 13.22 | 39.03 | 22.00 | 7.65 | **32.05** | | N/A | | | N/A | | | | | | | |
| HCPIMSC | 33.63 | 14.14 | 37.16 | 21.99 | 8.92 | 29.84 | 9.66 | 12.22 | 11.15 | | | | | N/A | | | N/A | |
| EEIMVC | 33.40 | 12.68 | 35.74 | 22.00 | 6.12 | 13.59 | 5.85 | 14.65 | 5.74 | 3.37 | 7.17 | 4.56 | | | | | | |
| LRGRIMVC | 33.89 | 11.58 | 34.72 | 21.64 | 7.24 | 30.12 | 9.26 | 13.03 | 12.56 | | N/A | | | | | | | |
| IMVCCBG | 40.72 | 14.13 | 39.36 | 22.03 | 8.22 | 31.15 | 11.78 | 13.70 | 12.68 | 7.69 | 13.81 | **8.65** | 74.79 | 79.01 | 77.30 | 56.76 | 57.41 | 60.19 |
| BGIMVSC | 23.60 | 6.29 | 24.20 | 19.01 | 0.30 | 22.83 | 7.34 | 10.41 | 7.76 | | N/A | | | N/A | | | N/A | |
| OSLFIMVC | 31.85 | 8.58 | 35.67 | 20.96 | 6.62 | 32.11 | 8.15 | 8.04 | 8.86 | 4.94 | 8.39 | 5.42 | 59.67 | 69.78 | 70.15 | 40.78 | 36.57 | 46.37 |
| NGSPCGL | 28.58 | 5.92 | 29.47 | 19.68 | 5.14 | 28.36 | 6.24 | 5.77 | 6.86 | | N/A | | | N/A | | | N/A | |
| PIMVC | 32.16 | 12.94 | 33.34 | 19.62 | **9.04** | 31.13 | 9.14 | 13.04 | 11.13 | 6.12 | 13.25 | 7.31 | | | | | | |
| PSIMVC | 37.05 | 12.64 | 38.56 | 20.27 | 8.26 | 29.02 | 10.14 | 11.78 | 10.98 | 5.45 | 10.27 | 6.00 | 71.85 | 74.66 | 76.75 | 53.56 | 56.45 | 57.45 |
| SAGL | 24.35 | 1.58 | 24.56 | 19.49 | 7.22 | 26.47 | 8.49 | 9.41 | 7.66 | 5.14 | 9.33 | 5.28 | 63.48 | 71.98 | 73.24 | 42.37 | 57.33 | 54.63 |
| HCLSCGL | 27.28 | 4.54 | 28.40 | 21.26 | 7.85 | 31.30 | 5.18 | 9.02 | 5.47 | 3.06 | 10.55 | 4.15 | | N/A | | | N/A | |
| Ours | **41.03** | **14.27** | **42.43** | **22.32** | 8.50 | 31.97 | **12.68** | **14.85** | **13.45** | **7.77** | **14.26** | 8.60 | **75.34** | **79.89** | **79.97** | **61.92** | **60.86** | **65.38** |

Table 8: Performance Comparison under Missing Ratio = 60%

| Method | BDGPFEA | | | NUSOBJECT | | | VGGFACEFIFTY | | | VGGFACEHUND | | | YOUTUBEFACE | | | FASHMINST | | |
|---|---|---|---|---|---|---|---|---|---|---|---|---|---|---|---|---|---|---|
| | ACC | NMI | PUR | ACC | NMI | PUR | ACC | NMI | PUR | ACC | NMI | PUR | ACC | NMI | PUR | ACC | NMI | PUR |
| LSIMVC | 27.16 | 6.51 | 27.16 | 20.90 | 8.10 | 31.21 | 7.17 | 9.53 | 7.42 | | | | | | | | | |
| GSRIMC | 35.53 | 13.29 | 36.33 | 20.47 | 7.78 | 30.87 | | N/A | | | N/A | | | | | | | |
| HCPIMSC | 31.63 | 11.96 | 32.75 | 22.40 | 8.01 | 29.70 | 9.14 | 10.34 | 10.45 | | | | | N/A | | | N/A | |
| EEIMVC | 32.90 | 10.35 | 34.02 | 22.14 | 8.05 | 14.01 | 5.49 | 13.73 | 5.39 | 3.35 | 6.42 | 4.26 | | | | | | |
| LRGRIMVC | 30.29 | 8.24 | 31.54 | 21.22 | 7.66 | 30.73 | 9.80 | 12.33 | 11.26 | | N/A | | | | | | | |
| IMVCCBG | 36.09 | 10.82 | 35.97 | 22.05 | 7.34 | 32.00 | 11.19 | 12.92 | 12.10 | 7.13 | 12.74 | 7.67 | 74.33 | 78.85 | 78.33 | **57.99** | 57.62 | **61.14** |
| BGIMVSC | 23.32 | 6.08 | 23.76 | 19.13 | 0.41 | 22.89 | 7.20 | 10.22 | 7.70 | | N/A | | | N/A | | | N/A | |
| OSLFIMVC | 31.64 | 8.24 | 36.46 | 20.27 | 4.76 | 30.14 | 6.54 | 5.74 | 7.17 | 4.01 | 6.62 | 4.54 | 61.89 | 68.37 | 69.18 | 40.18 | 33.56 | 43.47 |
| NGSPCGL | 27.24 | 4.23 | 27.73 | 18.75 | 3.33 | 26.43 | 6.03 | 6.25 | 6.66 | | N/A | | | N/A | | | N/A | |
| PIMVC | **37.10** | 11.39 | 38.08 | 19.78 | 8.08 | 30.92 | 8.98 | 12.08 | 10.92 | 5.85 | 12.57 | 6.94 | | | | | | |
| PSIMVC | 32.47 | 9.49 | 33.52 | 22.08 | 7.95 | 30.14 | 9.84 | 11.12 | 11.07 | 5.29 | 10.11 | 5.84 | 68.39 | 72.70 | 74.56 | 54.53 | 54.94 | 57.86 |
| SAGL | 23.11 | 1.29 | 23.36 | 19.39 | 6.76 | 25.98 | 6.15 | 9.22 | 6.85 | 4.04 | 10.27 | 4.64 | 63.39 | 72.16 | 74.18 | 42.37 | 58.76 | 54.56 |
| HCLSCGL | 20.80 | 1.46 | 20.80 | 20.42 | 7.74 | 31.19 | 4.57 | 8.30 | 5.08 | 2.96 | 10.45 | 4.01 | | N/A | | | N/A | |
| Ours | 36.13 | **12.09** | 38.13 | 21.99 | **8.16** | **32.15** | 11.65 | 13.83 | 12.53 | 7.14 | 13.18 | **7.88** | **74.98** | **80.49** | **79.68** | 57.95 | **59.49** | 60.94 |

# J   MORE RESULT COMPARISON ON OTHER MISSING RATIOS

To further reveal the ability of our SIIHPC to effectively tackle incomplete multi-view data, we also organize the comparison experiments under missing ratio = 40%, 60% and 80%. The results are reported in Table 7, 8 and 9 respectively. It can bee seen that SIIHPC still receives preferable clustering results in most cases, even under fairly high missing ratio (like 80%). There results illustrate that the proposed SIIHPC is able to effectively deal with incomplete multi-view data with diverse missing ratios.

Table 9: Performance Comparison under Missing Ratio = 80%

| Method | BDGPFEA | | | NUSOBJECT | | | VGGFACEFIFTY | | | VGGFACEHUND | | | YOUTUBEFACE | | | FASHMINST | | |
|---|---|---|---|---|---|---|---|---|---|---|---|---|---|---|---|---|---|---|
| | ACC | NMI | PUR | ACC | NMI | PUR | ACC | NMI | PUR | ACC | NMI | PUR | ACC | NMI | PUR | ACC | NMI | PUR |
| LSIMVC | 28.33 | 6.92 | 28.78 | 19.87 | 7.05 | 30.14 | 6.86 | 8.88 | 7.12 | | | | | | | | | |
| GSRIMC | 34.94 | **11.64** | 31.71 | 20.66 | 7.62 | 30.93 | | N/A | | | N/A | | | | | | | |
| HCPIMSC | 33.34 | 11.43 | 34.37 | 20.79 | 7.51 | 28.82 | 9.47 | 8.88 | 10.32 | | | | | N/A | | | N/A | |
| EEIMVC | 30.49 | 9.80 | 32.89 | 20.56 | 7.75 | 13.13 | 5.14 | 12.71 | 5.05 | 3.27 | 6.42 | 3.57 | | | | | | |
| LRGRIMVC | 26.84 | 3.97 | 27.45 | **21.83** | 6.91 | 29.37 | 9.04 | 11.78 | 11.83 | | N/A | | | | | | | |
| IMVCCBG | 35.56 | 9.13 | 34.01 | 21.04 | 7.37 | 30.53 | 10.59 | 11.95 | 11.43 | 6.52 | 11.73 | 7.17 | 69.99 | **78.24** | 75.56 | 55.75 | 56.05 | 59.50 |
| BGIMVSC | 20.88 | 1.45 | 21.32 | 19.07 | 2.05 | 22.93 | 6.81 | 10.18 | 7.28 | | N/A | | | | | | N/A | |
| OSLFIMVC | 30.37 | 8.97 | 34.64 | 18.91 | 4.16 | 30.00 | 5.49 | 4.56 | 6.31 | 3.42 | 5.33 | 3.87 | 60.79 | 68.28 | 68.36 | 42.22 | 36.14 | 43.60 |
| NGSPCGL | 27.59 | 5.67 | 27.94 | 17.80 | 2.11 | 24.38 | 6.30 | 6.22 | 6.85 | | N/A | | | N/A | | | N/A | |
| PIMVC | 32.46 | 9.57 | 32.46 | 19.28 | 7.62 | **31.88** | 8.54 | 11.62 | 11.88 | 5.57 | 11.98 | 6.61 | | | | | | |
| PSIMVC | 32.13 | 8.61 | 32.95 | 19.86 | 7.78 | 30.20 | 9.37 | 10.09 | 9.96 | 5.24 | 9.81 | 5.68 | 66.47 | 72.15 | 74.60 | 54.36 | 54.31 | 57.62 |
| SAGL | 27.56 | 5.23 | 30.80 | 17.58 | 6.16 | 26.25 | 5.44 | 9.18 | 7.58 | 3.85 | 8.36 | 4.36 | 61.37 | 71.57 | 72.47 | 44.53 | 53.67 | 55.01 |
| HCLSCGL | 27.49 | 3.85 | 27.80 | 19.74 | **7.82** | 29.57 | 4.98 | 8.02 | 5.31 | 3.25 | 9.33 | 3.89 | | N/A | | | N/A | |
| Ours | **36.79** | 10.71 | **37.91** | 21.38 | 7.19 | 31.75 | **11.20** | **13.17** | **12.03** | **6.60** | **12.12** | **7.37** | **70.40** | 77.78 | **75.72** | **66.15** | **60.96** | **66.15** |

## K  HYPER-PARAMETER SENSITIVITY

To explore the sensitivity of our SIIHPC to hyper-parameters $\lambda$ and $\beta$, we collect the clustering results under different combinations of $\lambda$ and $\beta$, and present them in Fig. 7. One can observe that for any fixed $\beta$, there is no significant fluctuation in the clustering performance. On the other hand, for any fixed $\lambda$, although the clustering performance is fluctuating with the change of $\beta$, it is still tolerable to some extent. Therefore, our SIIHPC is not significantly affected by $\lambda$ and $\beta$.

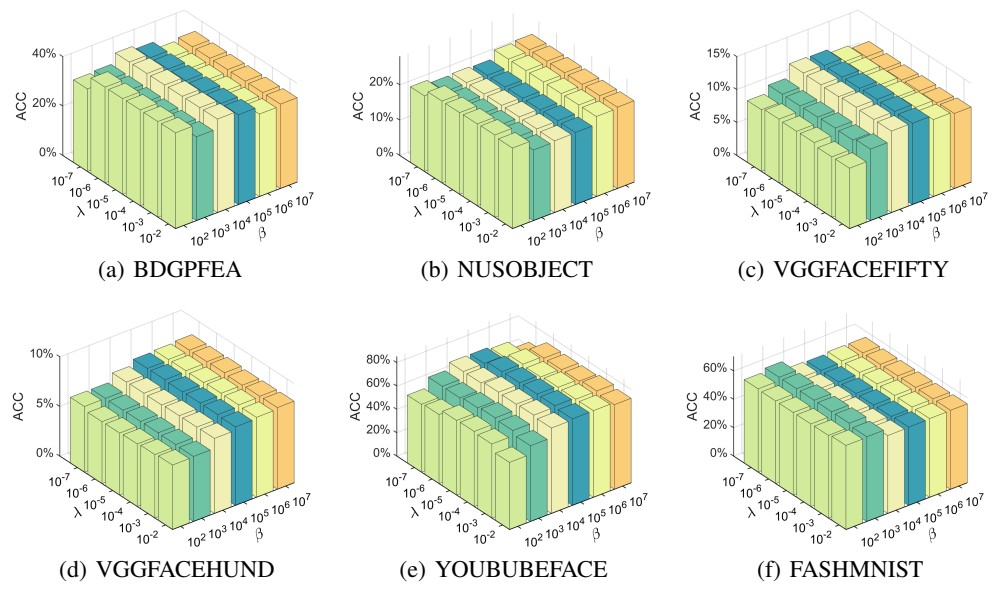

(a) BDGPFEA  (b) NUSOBJECT  (c) VGGFACEFIFTY

(d) VGGFACEHUND  (e) YOUBUBEFACE  (f) FASHMNIST

Figure 7: The clustering results of SIIHPC under different hyper-parameters.

## L  THE INFLUENCE OF HYBRID PROTOTYPE QUANTITIES

In this paper, the hybrid prototype quantities on each view are set as $[1k, 2k, 3k, 4k, 5k]$. We consider that too few prototype quantities are not adequately to exploit data features, while too many prototype quantities could lead to information redundancy and the increasing of running time. To explore the influence of hybrid prototype quantities on the clustering performance, we conduct multiple comparison experiments and collect the clustering results under $[1k, 2k]$, $[1k, 2k, 3k]$, $\cdots$ $[1k, 2k, \cdots, 8k]$ respectively, as shown in Table 10, 11 and 12 where SHPQ: the Setting of Hybrid Prototype Quantities. It can be seen that when $m_1 \sim m_S$ take $[1k, 2k]$, the obtained results are inferior than that $m_1 \sim m_S$ taking $[1k, 2k, \cdots, 5k]$, $[1k, 2k, \cdots, 6k]$ and $[1k, 2k, \cdots, 7k]$ in most instances. For instance, on VGGFACEFIFTY with missing ratio = 30%, 50% and 70% in ACC,

NMI and PUR, the results under $[1k, 2k]$ are lower than that under $[1k, 2k, \cdots, 5k]$, $[1k, 2k, \cdots, 6k]$ and $[1k, 2k, \cdots, 7k]$ by (1.16%, 1.39%, 1.28%, 0.77%, 0.86%, 0.77%, 1.15%, 1.45%, 0.96%), (1.35%, 1.64%, 1.48%, 0.74%, 0.85%, 0.58%, 1.24%, 1.44%, 1.07%) and (1.14%, 1.27%, 1.20%, 0.81%, 0.88%, 0.80%, 1.30%, 1.72%, 1.23%), respectively. There illustrate that too few prototype quantities could be not conductive to the improvement of clustering results. On the other hand, when $m_1 \sim m_S$ take $[1k, 2k, \cdots, 8k]$, the obtained results are also inferior than $m_1 \sim m_S$ taking $[1k, 2k, \cdots, 5k]$ and $[1k, 2k, \cdots, 6k]$ in most cases. For example, on FASHMINST with missing ratio = 30%, 50% and 70% in ACC, NMI and PUR, the results under $[1k, 2k, \cdots, 8k]$ are lower than that under $[1k, 2k, \cdots, 5k]$ and $[1k, 2k, \cdots, 6k]$ by (9.50%, 2.69%, 6.48%, 6.57%, 2.46%, 4.30%, 2.10%, 0.91%, 0.84%) and by (9.13%, 2.40%, 6.02%, 1.45%, 0.36%, 0.71%, 3.00%, 0.25%, 2.02%), respectively. Also, the running time (in second) under $[1k, 2k, \cdots, 8k]$ is also relatively higher than that under $[1k, 2k, \cdots, 5k]$ and $[1k, 2k, \cdots, 6k]$. In addition to these, combined with Table 2, we know that SIIHPC is able to beat most of comparison algorithms when $m_1 \sim m_S$ take $[1k, 2k, \cdots, 5k]$. Based on these, we hold that $m_1 \sim m_S$ taking $[1k, 2k, \cdots, 5k]$ is worth recommending in our SIIHPC.

Table 10: The Influence of Prototype Quantity on Performance (Results on BDGPFEA)

| SHPQ | 30% | | | | 50% | | | | 70% | | | |
|---|---|---|---|---|---|---|---|---|---|---|---|---|
| | ACC | NMI | PUR | Time | ACC | NMI | PUR | Time | ACC | NMI | PUR | Time |
| $m_1 \sim m_S = [1k, 2k]$ | 39.32 | 13.36 | 40.64 | 5.54 | 38.83 | 9.30 | 38.83 | 1.81 | 32.49 | 5.84 | 33.62 | 1.62 |
| $m_1 \sim m_S = [1k, 2k, 3k]$ | 35.11 | 11.83 | 38.21 | 8.74 | 38.59 | 10.47 | 38.59 | 1.94 | 34.50 | 8.22 | 35.35 | 1.87 |
| $m_1 \sim m_S = [1k, 2k, 3k, 4k]$ | 35.63 | 13.35 | 36.37 | 8.29 | 38.95 | 10.93 | 39.71 | 2.29 | 34.39 | 8.11 | 36.74 | 2.10 |
| $m_1 \sim m_S = [1k, 2k, \cdots, 5k]$ | 38.80 | 15.21 | 39.97 | 7.16 | 40.31 | 13.88 | 40.31 | 2.02 | 35.04 | 11.54 | 37.30 | 1.45 |
| $m_1 \sim m_S = [1k, 2k, \cdots, 6k]$ | 39.20 | 15.65 | 39.91 | 10.16 | 40.39 | 13.32 | 42.58 | 3.81 | 38.80 | 9.61 | 38.80 | 3.06 |
| $m_1 \sim m_S = [1k, 2k, \cdots, 7k]$ | 37.55 | 16.12 | 39.54 | 13.54 | 36.98 | 13.44 | 38.93 | 3.70 | 39.12 | 10.01 | 39.12 | 4.28 |
| $m_1 \sim m_S = [1k, 2k, \cdots, 8k]$ | 37.18 | 15.24 | 38.61 | 15.02 | 41.04 | 14.11 | 41.41 | 4.58 | 39.06 | 10.71 | 39.06 | 3.77 |

Table 11: The Influence of Prototype Quantity on Performance (Results on VGGFACEFIFTY)

| SHPQ | 30% | | | | 50% | | | | 70% | | | |
|---|---|---|---|---|---|---|---|---|---|---|---|---|
| | ACC | NMI | PUR | Time | ACC | NMI | PUR | Time | ACC | NMI | PUR | Time |
| $m_1 \sim m_S = [1k, 2k]$ | 11.36 | 13.52 | 12.16 | 60.74 | 11.54 | 13.62 | 12.44 | 61.48 | 10.03 | 11.90 | 11.00 | 48.74 |
| $m_1 \sim m_S = [1k, 2k, 3k]$ | 12.05 | 14.18 | 12.86 | 85.43 | 12.07 | 14.20 | 12.81 | 85.75 | 10.49 | 12.66 | 11.33 | 75.81 |
| $m_1 \sim m_S = [1k, 2k, 3k, 4k]$ | 12.08 | 14.36 | 12.90 | 115.63 | 11.99 | 14.27 | 12.87 | 115.57 | 11.14 | 13.16 | 12.01 | 107.57 |
| $m_1 \sim m_S = [1k, 2k, \cdots, 5k]$ | 12.52 | 14.91 | 13.44 | 112.19 | 12.31 | 14.48 | 13.21 | 113.03 | 11.18 | 13.35 | 11.96 | 109.47 |
| $m_1 \sim m_S = [1k, 2k, \cdots, 6k]$ | 12.71 | 15.16 | 13.64 | 214.75 | 12.28 | 14.47 | 13.02 | 211.99 | 11.27 | 13.34 | 12.07 | 215.13 |
| $m_1 \sim m_S = [1k, 2k, \cdots, 7k]$ | 12.50 | 14.79 | 13.36 | 263.73 | 12.35 | 14.50 | 13.24 | 262.95 | 11.33 | 13.62 | 12.23 | 257.53 |
| $m_1 \sim m_S = [1k, 2k, \cdots, 8k]$ | 12.61 | 15.02 | 13.47 | 343.71 | 12.41 | 14.62 | 13.27 | 352.80 | 11.44 | 13.51 | 12.31 | 353.56 |

Table 12: The Influence of Prototype Quantity on Performance (Results on FASHMINST)

| SHPQ | 30% | | | | 50% | | | | 70% | | | |
|---|---|---|---|---|---|---|---|---|---|---|---|---|
| | ACC | NMI | PUR | Time | ACC | NMI | PUR | Time | ACC | NMI | PUR | Time |
| $m_1 \sim m_S = [1k, 2k]$ | 55.22 | 59.34 | 59.54 | 137.94 | 54.90 | 58.55 | 60.45 | 139.56 | 56.23 | 59.16 | 60.48 | 162.99 |
| $m_1 \sim m_S = [1k, 2k, 3k]$ | 55.98 | 58.32 | 59.86 | 195.92 | 55.43 | 57.54 | 59.55 | 193.96 | 62.21 | 55.65 | 62.21 | 201.49 |
| $m_1 \sim m_S = [1k, 2k, 3k, 4k]$ | 56.02 | 59.30 | 60.11 | 242.65 | 62.14 | 56.38 | 62.26 | 251.98 | 59.60 | 53.82 | 62.19 | 255.66 |
| $m_1 \sim m_S = [1k, 2k, \cdots, 5k]$ | 61.24 | 59.52 | 62.69 | 215.01 | 62.51 | 60.22 | 64.64 | 213.51 | 60.59 | 58.77 | 63.18 | 213.26 |
| $m_1 \sim m_S = [1k, 2k, \cdots, 6k]$ | 60.87 | 59.23 | 62.23 | 371.20 | 57.39 | 58.12 | 61.05 | 376.81 | 61.49 | 58.11 | 64.36 | 383.28 |
| $m_1 \sim m_S = [1k, 2k, \cdots, 7k]$ | 57.95 | 59.24 | 61.86 | 451.67 | 55.66 | 58.74 | 59.95 | 453.26 | 61.24 | 60.20 | 64.35 | 450.35 |
| $m_1 \sim m_S = [1k, 2k, \cdots, 8k]$ | 51.74 | 56.83 | 56.21 | 540.36 | 55.94 | 57.76 | 60.34 | 522.42 | 58.49 | 57.86 | 62.34 | 534.14 |

## M STABILITY AND RELIABILITY

To highlight the stability and reliability of our SIIHPC's performance, we count the standard deviation (%) of clustering results under diverse missing ratios, as shown in Table 13 and Table 14.

As seen, even under diverse missing rations, our standard deviation is small, which illustrates that our SIIHPC is robust under varying data conditions.

Table 13: The Standard Deviation Results (%) (Part One)

| BDGPFEA | | | | | | | | |
|---|---|---|---|---|---|---|---|---|
| 30% | | | 50% | | | 70% | | |
| ACC | NMI | PUR | ACC | NMI | PUR | ACC | NMI | PUR |
| 0.25 | 0.16 | 0.37 | 0.24 | 0.02 | 0.24 | 0.32 | 0.01 | 0.06 |
| NUSOBJECT | | | | | | | | |
| 0.30 | 0.13 | 0.17 | 0.09 | 0.14 | 0.20 | 0.13 | 0.06 | 0.16 |
| VGGFACEFIFTY | | | | | | | | |
| 0.30 | 0.35 | 0.33 | 0.32 | 0.31 | 0.27 | 0.36 | 0.31 | 0.29 |
| VGGFACEHUND | | | | | | | | |
| 0.25 | 0.25 | 0.25 | 0.24 | 0.16 | 0.25 | 0.19 | 0.22 | 0.18 |
| YOUTUBEFACE | | | | | | | | |
| 2.46 | 1.07 | 1.58 | 2.09 | 1.30 | 1.88 | 2.55 | 0.82 | 1.67 |
| FASHMINST | | | | | | | | |
| 0.00 | 0.00 | 0.00 | 0.00 | 0.01 | 0.01 | 0.00 | 0.28 | 0.00 |

Table 14: The Standard Deviation Results (%) (Part Two)

| BDGPFEA | | | | | | | | |
|---|---|---|---|---|---|---|---|---|
| 40% | | | 60% | | | 80% | | |
| ACC | NMI | PUR | ACC | NMI | PUR | ACC | NMI | PUR |
| 0.18 | 0.06 | 0.09 | 0.09 | 0.01 | 0.13 | 0.06 | 0.06 | 0.06 |
| NUSOBJECT | | | | | | | | |
| 0.18 | 0.19 | 0.12 | 0.52 | 0.09 | 0.09 | 0.18 | 0.07 | 0.14 |
| VGGFACEFIFTY | | | | | | | | |
| 0.32 | 0.23 | 0.31 | 0.47 | 0.33 | 0.29 | 0.51 | 0.35 | 0.41 |
| VGGFACEHUND | | | | | | | | |
| 0.22 | 0.24 | 0.19 | 0.16 | 0.19 | 0.15 | 0.17 | 0.20 | 0.16 |
| YOUTUBEFACE | | | | | | | | |
| 2.07 | 0.83 | 1.47 | 2.97 | 0.82 | 1.50 | 2.42 | 1.26 | 2.18 |
| FASHMINST | | | | | | | | |
| 0.48 | 0.00 | 0.01 | 0.01 | 0.00 | 0.01 | 0.02 | 0.01 | 0.02 |

## N    WEIGHTS OF HYBRID PROTOTYPE QUANTITIES

Instead of a single prototype quantity for all views, we generate a group of hybrid prototype quantities for each view to flexible capture features. Further, we associate a weight variable for each prototype quantity on each view to adaptively balance their contribution. To verify that it learns different weights for prototype quantities, we visualize these weights, as shown in Fig. 8. It is easy to see that on each view, each prototype quantity indeed enjoys a different weight, which makes it more flexibly combine these prototype quantities to extract features.

## O    PERFORMANCE CHANGE W.R.T MISSING RATIO

To illustrate that the clustering performance of our proposed SIIHPC is not significantly affected by the missing ratio, we collect the clustering results under multiple missing ratios, and report them in Fig. 9. As seen, along with the increasing of missing ratio, the clustering performance in most cases is gradually decreasing. This is mainly because the increasing of missing ratio leads to a gradual reduction in available information and the clustering performance is accordingly reducing. Despite

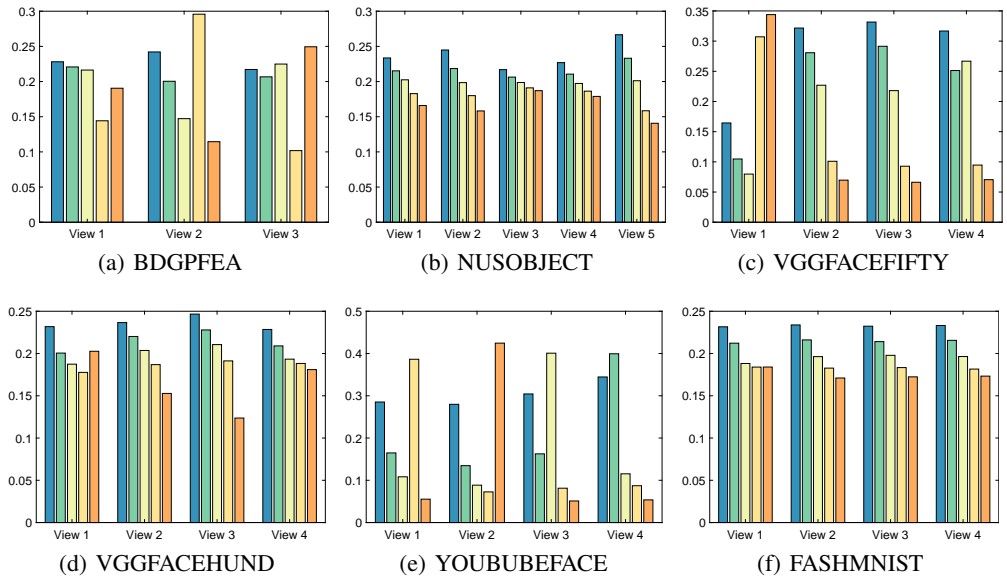

Figure 8: The learned weights of prototype quantities on each view.

all this, from Fig. 9, one can observe that our clustering performance does not fluctuate drastically with the missing ratio.

## P    LIMITATION AND FUTURE WORK

We aggregate the information of graphs with diverse scales by stacking them and performing SVD to construct splicing eigenvectors, which could weaken the contributions of certain graphs or certain eigen-components. Other ingenious fusion schemes are worth further exploration. Additionally, we utilize the cosine tool to do similarity measure, some other measure approaches can be further studied in the future.

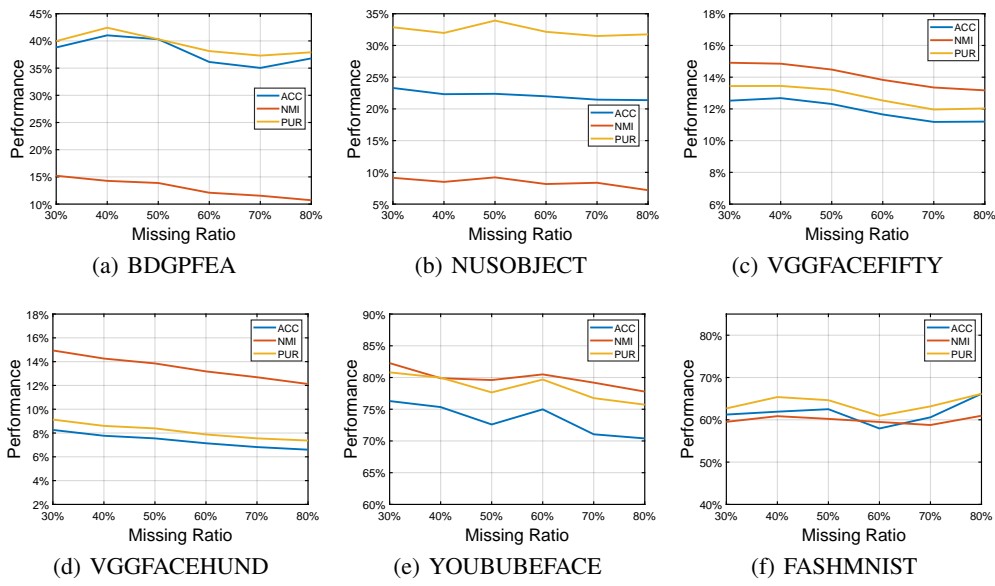

Figure 9: The change in clustering performance w.r.t the missing ratio.

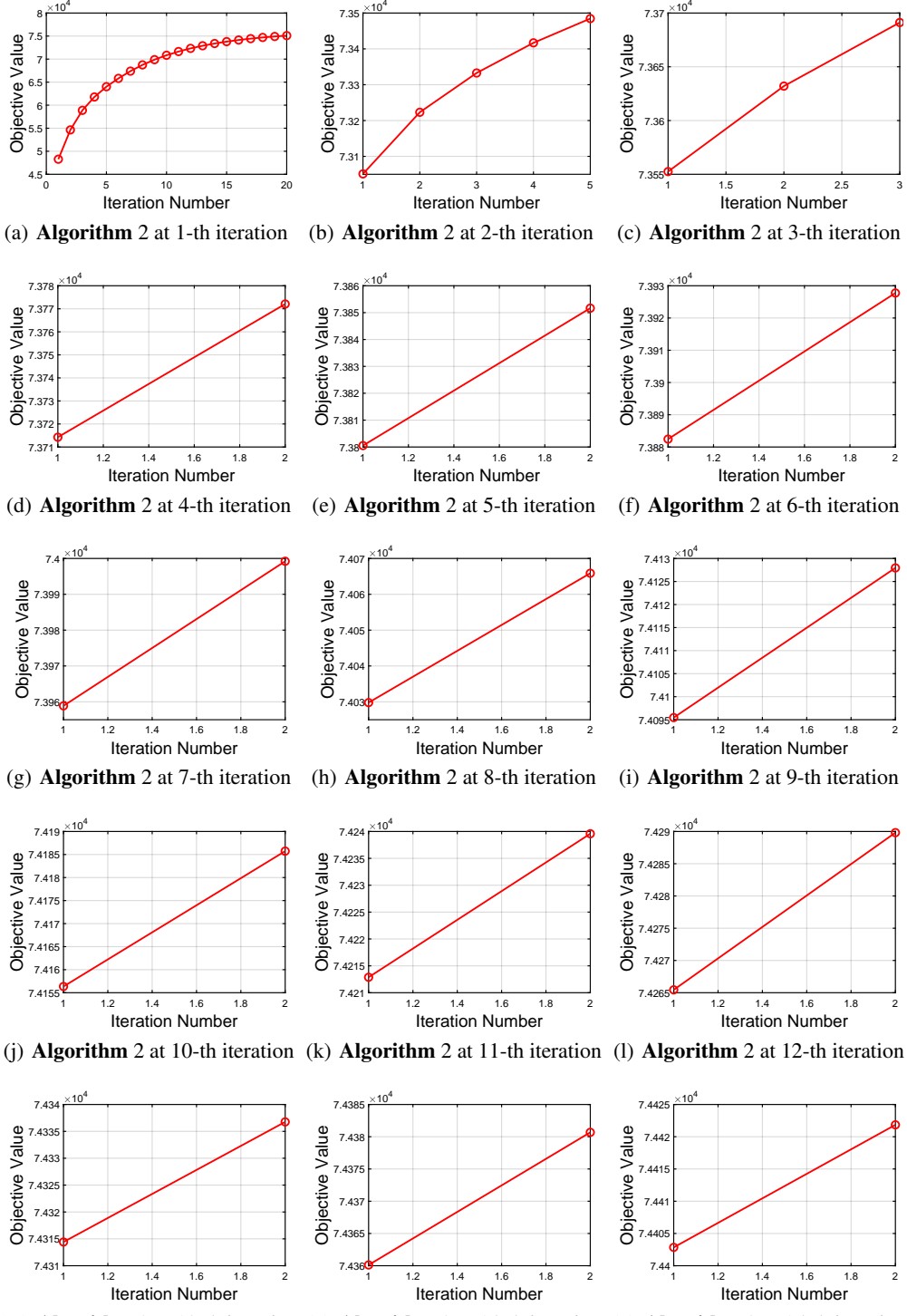

Figure 10: The value change of function $g$ on YOUBUBEFACE.

