# OpenReview forum: "Simple yet Effective Incomplete Multi-view Clustering: Similarity-level Imputation and Intra-view Hybrid-group Prototype Construction"
_ICLR.cc/2025/Conference — ICLR 2025 Spotlight_

### Official Review · Reviewer_Gpnt · 2024-10-30

**Soundness:** 4
**Presentation:** 4
**Contribution:** 3
**Rating:** 8
**Confidence:** 5

**Summary:**

This paper presents an incomplete multi-view clustering (IMVC) algorithm, SIIHPC. To utilize the latent valuable information of incomplete samples, it successfully separates out the observed parts via bipartition learning transformation and relaxes traditional non-negative constraints via sample regularization. It introduces the learnable consensus graphs to provide unified structure, and based on the relationship between consensus graphs and exclusive similarities it imputes the incomplete part at the similarity level. To get rid of the limitations of single prototype quantity, it assigns a group of hybrid prototype quantities for each view, and performs spectral grouping on the resulting multi-scale graphs to generate the cluster labels. Afterwards, it gives a four-step updating scheme to minimize the formulated objective. Overall, the overall organization of this manuscript is clear and easy to follow.

**Strengths:**

1. The motivation is good. Incorporating incomplete samples into the construction of bipartite similarity enhances the representation of relationships between different entities.   Diverse prototype quantities facilitate the accurate characterization of different views.

2. The designed auxiliary function is interesting. It equivalently solves one quadratic programming problem with orthogonal constraints and meanwhile is demonstrated to be with monotonic-increasing properties.

**Weaknesses:**

1. In the paper, the hyper-parameters $\lambda$ and $\beta$ are adjusted in $[10^{-7}, 10^{-6}, \cdots, 10^{-2} ]$  and $[10^{2}, 10^{3}, \cdots, 10^{7} ]$ respectively. The reasons for setting parameters are not given.

2. Similar to the analysis about Table 3, some reasons for the sub-optimal results in Table 2 should be added.

3. Descriptions about the theoretical space overhead should be more in-depth.

**Questions:**

1. As descripted, PSIMVC is also based on prototype, what are the differences between the proposed SIIHPC and PSIMVC? Specially, combined with Table 3, PSIMVC can consume less time and/or memory.

2. Figure 4 show that at each iteration of Algorithm 2, the objective of function $g$ is increasing while at different iterations of Algorithm 2, it doesn't seem like this, such as at the $3$-th, $4$-th, $5$-th iteration respectively.

3. After transformation, why is the complexity about $\mathbf{H}_{v,s}$ reduced from square to linear?

4. When adopting the prototypes constructed by elicitation sampling, what is the performance of SIIHPC?

---

> ### Author Response · Authors · 2024-11-24
> **Reply for Reviewer Gpnt (1)**
>
> We extend our sincere gratitude to Reviewer  Gpnt for his/he valuable feedback and guidance in revising this paper. We have thoroughly addressed each concern raised and are hopeful that all issues have been resolved satisfactorily.
>
> **Q1:** In the paper, the hyper-parameters $\lambda$ and $\beta$ are adjusted in $[10^{-7}, 10^{-6}, \cdots, 10^{-2} ]$  and $[10^{2}, 10^{3}, \cdots, 10^{7} ]$ respectively. The reasons for setting parameters are not given.
>
> **A1:** Thanks for your constructive comment.  The hyper-parameters $\lambda$ and $\beta$ aim at fine-tuning  $\left\||   \mathbf{G}_s
>     \right\|| _F^2 $  and $\left\||  \mathbf{A} \right\|| _F^2$ respectively. Note that $\mathbf{G} _s \in \mathbb{R}^{m _s \times n}$ and $\mathbf{A} \in \mathbb{R}^{V \times S}$  where $m _s$, $n$, $V$ and $S$ denote the numbers of  prototypes at the $s$-th group, samples, views and prototype quantity candidates, respectively. Generally, $V \leq m_s $ and $S \ll n$. Therefore, we have that the size of $\mathbf{G}_s$ is much greater than that of $\mathbf{A}$. Besides, combined with the feasible regions, we have that the absolute values of the elements in $\mathbf{G} _s$ and $\mathbf{A}$ are all within the range $[0,1]$, Therefore, we can get that $\left\|| \mathbf{G} _s   \right\|| _F^2 $ is much larger than $\left\|| \mathbf{A}  \right\|| _F^2 $. Based on these, we search $\lambda$ within a small range while searching $\beta$ within a relatively large range.
>
> These have been added in Section A of the appendix. Please check it.
>
>
> **Q2:** Similar to the analysis about Table 3, some reasons for the sub-optimal results in Table 2 should be added.
>
> **A2:**  There exist a small amount  of sub-optimal results in some cases against the algorithms HCPIMSC, LRGRIMVC, IMVCCBG and PIMVC. Possible reasons could be that HCPIMSC exploits the high-order correlation between views via low-rank tensor learning and utilizes  a hyper-graph regularizer to capture geometrical structure  between samples; LRGRIMVC adopts adaptive graph embedding and rank constraint to extract view features, and combines the global structure,  relevant information within views and latent information across views together to construct the similarity relationship; IMVCCBG refines  consistency  view  features through consensus anchors and employs orthogonality learning to highlight the discrimination of potential sub-spaces;  PIMVC reformulates the projection learning of each view to alleviate the information imbalance and designs a graph penalty term to extract the geometric structure embedded into original data.
>
> **Q3:**   Descriptions about the theoretical space overhead should be more in-depth.
>
> **A3:**   The space overhead of Algorithm 2  mainly comes from $\\{ \mathbf{H} _{v,s} \\} _{v=1,s=1}^{V,S}$,  $\\{ \mathbf{Q} _{v,s} \\} _{v=1,s=1}^{V,S}$, $\\{ \mathbf{G} _{s} \\} _{s=1}^{S}$ and $\mathbf{A}$. Considering that their sizes are $d _v \times m _s$, $m _s \times n$, $m _s \times n$ and $V \times S$  respectively where $d _v$, $m _s$, $n$, $V$, $S$ denote the feature dimension on the $v$-th view,  the number of the $s$-th group prototypes, the number of samples, the number of views and the number of prototype quantity candidates,    we have that storing them requires $\mathcal{O}(\sum _{v=1}^{V} \sum _{s=1}^{S} d _v m _s)$, $\mathcal{O}(n \sum _{s=1}^{S} m _s)$, $\mathcal{O}(n \sum _{s=1}^{S} m_s  )$ and $\mathcal{O}(VS)$ memory overhead respectively. Therefore, the total space overhead is $\mathcal{O}(\sum _{v=1}^{V} \sum _{s=1}^{S} d _v m _s  +  n \sum _{s=1}^{S} m _s )$. Note that $d _v$ is a constant and not related to $n$ and that $m_s$ is generally far smaller than $n$, we can obtain that the space complexity is $\mathcal{O}(n)$, which is linear with respect to the number of samples.

---

> > ### Author Response · Authors · 2024-11-24
> > **Reply for Reviewer Gpnt (2)**
> >
> > **Q4:** As descripted, PSIMVC is also based on prototype, what are the differences between the proposed SIIHPC and PSIMVC? Specially, combined with Table 3, PSIMVC can consume less time and/or memory.
> >
> > **A4:**  PSIMVC constructs unified prototypes using a group of projections, and reduces the features of all prototypes to the same one dimension space. Due to involving only one prototype matrix, it in some cases  requires less time and/or memory.  However, it does not take into account the latent useful information of incomplete samples.  The unpairing of observed samples degrades the quality of generated unified affinity. Besides, the feature dimension unification inevitably causes  the loss of view information.  Also, the constructed unified prototypes are not conducive to  the extraction of complementary features between views.   Different from it, our SIIHPC utilizes view-specific prototypes to  extract features on each respective view and meanwhile integrates the contributions of incomplete samples into the  measure of similarity.  Further, our SIIHPC introduces a set of hybrid  prototype quantities for each view so as to adaptively form representations according to the characteristics of each view itself.
> >
> >
> > **Q5:**  Figure 4 show that at each iteration of Algorithm 2, the objective of function $g$ is increasing while at different iterations of Algorithm 2, it doesn't seem like this, such as at the $3$-th, $4$-th, $5$-th iteration respectively.
> >
> > **A5:**  The reason for this is that $\mathbf{H} _{v,s}$ and  $\mathbf{G} _{s}$ as well as  $\mathbf{Q} _{v,s}$ are all changing when Algorithm 2 is at different iterations.    The objective value of function $g$ at the initial iteration is influenced by  $\mathbf{H} _{v,s}$, $\mathbf{G} _{s}$ and  $\mathbf{Q} _{v,s}$.  Accordingly, under different iterations of Algorithm 2,  the objective value of function $g$ does not guarantee to be monotonous.     Under the same one iteration of Algorithm 2, the function $g$ is monotonically increasing, which has been proven in theory and experiments.  Besides, kindly note that the possible non-monotonicity of $g$ under different iterations of Algorithm 2 is nothing to do with the convergence of SIIHPC since   the function $g$ is designed by omitting some items unrelated to $\mathbf{H} _{v,s}$  when optimizing the sub-problem $\mathbf{H} _{v,s}$.
> >
> >
> > **Q6:**  After transformation, why is the complexity about $\mathbf{H}_{v,s}$ reduced from square to linear?
> >
> > **A6:** The solution of $\mathbf{H}_{v,s}$ can be acquired by Algorithm 1,  which involves the construction of function $g$. According to the definition of $g$, equivalently,  it needs to compute $\mathbf{L} _v$ and $\mathbf{P} _{v,s}$. Due to  $\mathbf{D} _v$, $\mathbf{W} _v$, $\mathbf{G} _s$, $\mathbf{Q} _{v,s}$ and $\mathbf{M} _v$ being in  $d _v \times n$, $n \times n _v$, $m_s \times n$, $m _s \times n$ and $n \times (n-n_v)$,   constructing  $\mathbf{L} _v$ and $\mathbf{P} _{v,s}$ will take  $\mathcal{O}(d _v n n _v + d _v^2 n)$ and $\mathcal{O}(d _v n n _v + m _s n (n-n _v) + d _v n m _s)$.  Generally, the feature dimension $d_v$ is greater than the prototype quantity $m_s$. Therefore, we have that the computational complexity of solving $\mathbf{H} _{v,s}$ is at least  $\mathcal{O}(d _v n n_v + d_v^2 n+  m_s n^2   + d_v n m_s )$.   Before transformation, therefore, the computational complexity is square with respect to the number of samples $n$.
> >
> > After transformation, $\mathbf{L} _v$ and $\mathbf{P} _{v,s}$ can be equivalently calculated by  $(\mathbf{D} _v \odot \mathbf{B} _v) \cdot (\mathbf{D} _v \odot \mathbf{B} _v)^{\top}$ and  $(2\mathbf{D} _v \odot \mathbf{B} _v) \cdot( \mathbf{G} _s - \mathbf{Q} _{v,s} \odot \mathbf{C} _v )^{\top}$ respectively where the sizes of $\mathbf{B} _v$ and $\mathbf{C} _v$ are in $d _v \times n$ and $m _s \times n$.  At this point, the computational complexity is  $\mathcal{O}(d _v n + d _v^2 n + d _v m _s n)$, which is linear with respective to $n$.   So, the complexity about $\mathbf{H} _{v,s}$ is reduced from square to linear after transformation.

---

> > > ### Author Response · Authors · 2024-11-24
> > > **Reply for Reviewer Gpnt (3)**
> > >
> > > **Q7:** When adopting the prototypes constructed by elicitation sampling, what is the performance of SIIHPC?
> > >
> > > **A7:** We organize relevant experiments, and report the comparison results in the following table where 'ES' denotes the clustering results based on the prototypes constructed by elicitation sampling.
> > >
> > >
> > > |      |           |           |           |           |  BDG. |           |           |           |           |
> > > |:----:|:---------:|:---------:|:---------:|:---------:|:--------:|:---------:|:---------:|:---------:|:---------:|
> > > |      |           |    30\%   |           |           |   50\%   |           |           |    70\%   |           |
> > > |      |    ACC    |    NMI    |    PUR    |    ACC    |    NMI   |    PUR    |    ACC    |    NMI    |    PUR    |
> > > |  ES  |   30.92   |    5.52   |   32.78   | **42.33** |   13.62  | **43.50** | **37.56** |   10.96   |   36.60   |
> > > | Ours | **38.80** | **15.21** | **39.97** |   40.31   | **13.88** |   40.31   |   35.04   | **11.54** | **37.30** |
> > >
> > >
> > >
> > >
> > >
> > > |      |           |          |           |           | NUS. |           |           |          |           |
> > > |:----:|:---------:|:--------:|:---------:|:---------:|:---------:|:---------:|:---------:|:--------:|:---------:|
> > > |      |           |   30\%   |           |           |    50\%   |           |           |   70\%   |           |
> > > |      |    ACC    |    NMI   |    PUR    |    ACC    |    NMI    |    PUR    |    ACC    |    NMI   |    PUR    |
> > > |  ES  |   22.22   |   8.16   |   32.05   |   20.48   |    7.59   |   30.84   |   19.28   |   5.86   |   30.32   |
> > > | Ours | **23.30** | **9.14** | **32.87** | **22.38** |  **9.21** | **33.92** | **21.46** | **8.36** | **31.49** |
> > >
> > >
> > >
> > >
> > >
> > > |      |           |           |           |           | VGY. |           |           |           |           |
> > > |:----:|:---------:|:---------:|:---------:|:---------:|:------------:|:---------:|:---------:|:---------:|:---------:|
> > > |      |           |    30\%   |           |           |     50\%     |           |           |    70\%   |           |
> > > |      |    ACC    |    NMI    |    PUR    |    ACC    |      NMI     |    PUR    |    ACC    |    NMI    |    PUR    |
> > > |  ES  |    9.22   |   10.76   |    9.93   |    7.83   |     9.11     |    8.58   |    9.31   |   11.25   |   10.24   |
> > > | Ours | **12.52** | **14.91** | **13.44** | **12.31** |   **14.48**  | **13.21** | **11.18** | **13.35** | **11.96** |
> > >
> > >
> > > |      |          |           |          |          | VGD. |          |          |           |          |
> > > |:----:|:--------:|:---------:|:--------:|:--------:|:-----------:|:--------:|:--------:|:---------:|:--------:|
> > > |      |          |    30\%   |          |          |     50\%    |          |          |    70\%   |          |
> > > |      |    ACC   |    NMI    |    PUR   |    ACC   |     NMI     |    PUR   |    ACC   |    NMI    |    PUR   |
> > > |  ES  |   7.57   |   14.26   |   8.45   |   5.25   |    10.27    |   5.88   |   4.48   |    9.25   |   5.16   |
> > > | Ours | **8.26** | **14.94** | **9.13** | **7.55** |  **13.85**  | **8.39** | **6.82** | **12.69** | **7.55** |
> > >
> > >
> > >
> > >
> > >
> > > |      |           |           |           |           | YOU. |           |           |           |           |
> > > |:----:|-----------|-----------|-----------|-----------|-------------|-----------|-----------|-----------|-----------|
> > > |      |           | 30\%      |           |           | 50\%        |           |           | 70\%      |           |
> > > |      | ACC       | NMI       | PUR       | ACC       | NMI         | PUR       | ACC       | NMI       | PUR       |
> > > | ES   | 74.64     | 78.46     | 79.71     | 70.08     | 71.85       | 73.75     | 63.56     | 66.32     | 68.99     |
> > > | Ours | **76.29** | **82.27** | **80.81** | **72.60** | **79.60**   | **77.65** | **71.05** | **79.19** | **76.75** |
> > >
> > >
> > >
> > >
> > > |      |           |           |           |           | FAS. |           |           |           |           |
> > > |:----:|:---------:|:---------:|:---------:|:---------:|:---------:|:---------:|:---------:|:---------:|:---------:|
> > > |      |           |    30\%   |           |           |    50\%   |           |           |    70\%   |           |
> > > |      |    ACC    |    NMI    |    PUR    |    ACC    |    NMI    |    PUR    |    ACC    |    NMI    |    PUR    |
> > > |  ES  |   46.55   |   42.23   |   47.06   |   39.04   |   37.73   |   40.76   |   34.07   |   36.05   |   36.84   |
> > > | Ours | **61.24** | **59.52** | **62.69** | **62.51** | **60.22** | **64.64** | **60.59** | **58.77** | **63.18** |
> > >
> > >
> > >  As seen, our SIIHPC receives preferable results in most cases, illustrating that our prototype generation strategy is desirable.

---

> > > > ### Author Response · Authors · 2024-11-28
> > > > **Reply for Reviewer Gpnt**
> > > >
> > > > Dear reviewer Gpnt,
> > > >
> > > > We deeply appreciate the time you took to provide your thoughtful and profound  feedback, and sincerely hope your concerns have been addressed in our response. If you have any additional suggestions, please feel free to contact us. We would be very happy to engage in a discussion with you. We understand that your time is valuable and appreciate your willingness to participate in this process.
> > > >
> > > >
> > > >
> > > > Best regards,
> > > >
> > > > Authors of paper 778

---

> > > > > ### Comment · Reviewer_Gpnt · 2024-11-28
> > > > >
> > > > > Dear Authors:
> > > > >
> > > > > Many thanks for your reminder and efforts.
> > > > >
> > > > > After checking the rebuttal, I believe the issues have been addressed. However, when I re-reading the paper, I think some details should be further completed, e.g., some notations should be explained more thoroughly in the main body; The initialization of $H_{v,s}$ seems unclear.
> > > > >
> > > > > Overall, this paper is well-presented, and I appreciate the authors' efforts. I will be raising my score accordingly.
> > > > >
> > > > > Best regards.

---

> > > > > > ### Author Response · Authors · 2024-11-28
> > > > > > **Reply for Reviewer Gpnt**
> > > > > >
> > > > > > Dear reviewer Gpnt,
> > > > > >
> > > > > > Thanks for recognizing our contributions.
> > > > > >
> > > > > > We will  in the main body provide more thorough explanations about the notations. For $\mathbf{H}_{v,s}$, we initialize it by utilizing k-means to generate a matrix with the same size  and  then doing orthogonalization on it.  We will elaborate more on this in the next verison.
> > > > > >
> > > > > > Sincerely thank you for the valuable time and profound suggestions.
> > > > > >
> > > > > >
> > > > > >
> > > > > >
> > > > > > Best regards,
> > > > > >
> > > > > > Authors of paper 778

---

### Official Review · Reviewer_yVBy · 2024-11-03

**Soundness:** 3
**Presentation:** 3
**Contribution:** 3
**Rating:** 8
**Confidence:** 5

**Summary:**

This paper proposes a new incomplete multi-view clustering method, which reconstructs similarity relationships through the original sample form to capture the intrinsic information of all views. By relaxing traditional non-negative constraints, the method enables more flexible similarity characterizations. Additionally, a hybrid prototype set is introduced for each view, facilitating view-specific feature extraction and contributing to a comprehensive consensus graph. An innovative auxiliary function with monotonic properties is designed to solve the optimization problem effectively. Experimental results on various incomplete multi-view datasets demonstrate the robust clustering performance of SIIHPC.

**Strengths:**

Paper Strengths
1.The paper is well-organized and easy to follow. The intuition is clearly discussed in the Introduction.
2.The proposed method performs exhaustive experimental analyses, comparing a wide array of clustering algorithms, thereby strengthening the credibility of the results.

**Weaknesses:**

Paper Weaknesses
1.The process for constructing the consensus graph $G$ is not clearly explained, particularly in relation to $G_s$.
2.Does the dataset include both images and text? Please specify the composition of each view within the dataset.
3.It would be helpful to discuss how Similarity Level Imputation and Hybrid-group Prototypes specifically contribute to improving clustering performance in Ablation section.

**Questions:**

1.The experimental setup lacks sufficient detail, particularly regarding the range and selection criteria of hyperparameters.
2.The Symbol Summary in the Appendix does not cover all symbols used throughout the paper, which may hinder clarity for readers. It is recommended to include all symbols and provide detailed explanations for each variable to improve readability and understanding of the methodology.

---

> ### Author Response · Authors · 2024-11-24
> **Reply for Reviewer yVBy (1)**
>
> We are very grateful for Reviewer yVBy constructive comments and guidance on the revision of this paper. We have carefully addressed all concerns, and truly hope that all issues have been resolved.
>
> **Q1:** The process for constructing the consensus graph $\mathbf{G}$ is not clearly explained, particularly in relation to  $\mathbf{G}_s$.
>
> **A1:**  $\mathbf{G}$ denotes a learnable consensus graph that is shared for all views.   (Kindly note that $\mathbf{H}_v^{\top} \mathbf{D}_v \mathbf{W}_v \mathbf{W}_v^{\top}$ denotes the observed similarity on view $v$.) $\mathbf{G}_s$ is the form of $\mathbf{G}$ under $s$-th prototype quantity candidate, and is constructed by Eq. (7) in the manuscript.
>
> **Q2:**  	Does the dataset include both images and text? Please specify the composition of each view within the dataset.
>
> **A2:**    The more detailed descriptions about the public multi-view datasets utilized in experiments are as follows.
>
> BDGPFEA is a genomic text sequence dataset, and consists of 2500 samples and  3 views. The feature dimensions on views are 1000, 500, and 250 respectively. The number of clusters is 5.
>
> NUSOBJECT is an object image dataset, and consists of 6251 samples and 5 views. The feature dimensions on views are  129, 74, 145, 226 and 65, respectively.  The number of clusters is 10.
>
> VGGFACEFIFTY is a face  dataset with 16936 samples.  The number of views is 4, and the feature dimensions are 944, 576, 512 and 640, respectively.  There are  50 clusters.
>
> VGGFACEHUND is also a fact dataset. It contains 36287 samples. The feature dimensions on 4 views are 512, 576, 640 and 944, respectively. There are  100 clusters.
>
> YOUTUBEFACE is an image dataset collected from YouTube website.  There are  63896 samples and 4 views totally.   The feature dimensions are 640, 944, 576 and 512 respectively.   The number of clusters is 20.
>
> FASHMINST is a fashion product database with 70000 samples and 4 views.  The feature dimensions are 576, 512, 944 and 640 respectively. The number of clusters is 10.
>
>
> **Q3:** It would be helpful to discuss how Similarity Level Imputation and Hybrid-group Prototypes specifically contribute to improving clustering performance in Ablation section.
>
> **A3:**   The more discussion about the effect of similarity level imputation and hybrid-group prototypes is as follows.
>
> For the similarity level imputation,  it  recovers the incomplete parts caused by missing samples  on current view through utilizing the representations of the same object on other views. It  alleviates the cluster imbalance caused by the unpairing of observed samples to improve the clustering performance. Also, it enables information from different views to interact mutually during the similarity forming phase  so as to generate more accurate similarity relationship.
>
> For the hybrid-group prototypes, it can utilize the learnable balance factors to automatically adjust the contributions of prototype quantities   according to the characteristics of each view itself, and thereby flexibly extracts view features and accordingly brings clustering performance improvement.
>
> Additionally, these two parts are in one common learning framework, which makes them  facilitate each other toward a mutual strengthening direction.

---

> ### Author Response · Authors · 2024-11-24
> **Reply for Reviewer yVBy (2)**
>
> **Q4:** The experimental setup lacks sufficient detail, particularly regarding the range and selection criteria of hyper-parameters.
>
> **A4:**   The parameter selection criteria is as follows.
>
> For $\lambda$ and $\beta$, we observe that they are associated with $\left\|| \mathbf{G}_s \right\|| _F^2 $  and $\left\||  \mathbf{A} \right\|| _F^2 $ respectively and the sizes of $\mathbf{G} _s$ and $\mathbf{A}$ are $m _s \times n$ and ${V \times S}$.  In general, the number of views $V$ is  less than or equal to the number  of  prototypes at the $s$-th group  $m_s$; the number of prototype quantity candidates $S$ is far less than the number of samples $n$.  Consequently, we can obtain that the size of $\mathbf{A}$ is fare less than that of $\mathbf{G}_s$. Beyond that, in conjunction with the feasible region constrains,  we can get that the element absolute values of  $\mathbf{A}$ and $\mathbf{G}_s$ are all in $0 \sim 1$.  As a result, we have that   $\left\|| \mathbf{A} \right\||_F^2$ is far less than  $\left\||  \mathbf{G} _s \right\|| _F^2$.  So, we fine-tune $\beta$,  which is related to	$\left\|| \mathbf{A}  \right\|| _F^2$, in a large range while fine-tuning $\lambda$ in a small range. These reasons about hyper-parameter setting are added in Section A of the appendix.  Please check it.
>
>
> For the hybrid prototype quantities on each view, we set them as $[1,2,\cdots,5]k$ where $k$ is the number of clusters.  The reason for this is that through experiments we find  too few prototype quantities are not adequately to exploit data features, while too many prototype quantities lead to information redundancy and the increasing of running time. Therefore, we set them as $[1,2,\cdots,5]k$ in all experiments.  More detailed exploration please refer to 'the influence of hybrid prototype quantities' section in Appendix.
>
> **Q5:** 	It is recommended to include all symbols in Symbol Summary and provide detailed explanations for each variable to improve readability and understanding of the methodology.
>
> **A5:**  Thanks! We have carefully modified the Symbol Summary section, and added more symbols and descriptions. Please check it.

---

> > ### Author Response · Authors · 2024-11-28
> > **Reply for Reviewer yVBy**
> >
> > Dear reviewer yVBy,
> >
> > We deeply appreciate the time you took to provide your thoughtful and profound  feedback, and sincerely hope your concerns have been addressed in our response. If you have any additional suggestions, please feel free to contact us. We would be very happy to engage in a discussion with you. We understand that your time is valuable and appreciate your willingness to participate in this process.
> >
> >
> >
> > Best regards,
> >
> > Authors of paper 778

---

### Official Review · Reviewer_3rbh · 2024-11-03

**Soundness:** 3
**Presentation:** 3
**Contribution:** 3
**Rating:** 8
**Confidence:** 5

**Summary:**

In this manuscript, the authors address the challenge of incompleteness due to missing samples by restoring the missing data at the similarity level. They propose a method that employs a hybrid approach to extract data representations, utilizing multiple quantities of prototypes for each individual view rather than relying on a single quantity across all views. This approach effectively resolves the misalignment of observed samples and incorporates potentially useful information from the missing samples into the bipartition similarity. Additionally, the authors balance contributions from different views while defining overall similarity based on the intrinsic characteristics of each view. These goals are effectively realized within a cohesive learning framework. The proposed method's effectiveness is convincingly demonstrated through experimental results on six public datasets with varying missing ratios.

**Strengths:**

1. The manuscript presents a logically structured approach, with Figure 1 providing a clear and intuitive overview of the framework.

2.  The authors utilize a variety of missing ratios and metrics in their experiments to evaluate clustering performance effectively.

3. The proposed SIIHPC demonstrates favorable memory and computational efficiency.

**Weaknesses:**

1. There is an issue with the manuscript's organization where the title of Section 5.5 is found to be identical to that of Section G in the Appendix.

2.  Experiments exhibit that some comparison methods do not function properly on VGGFACEHUND, YOUTUBEFACE and FASHMINST. However, there is a lack of theoretical explanations to substantiate these observations.

3. The space complexity of Algorithm 2 is not described enough.

4. The manuscript lacks definitions or explanations for certain notations used throughout the text. For instance, the notation $()_{+}$ in (11), N/A in Table 2 and Table 3  are not clarified.

**Questions:**

1. Is the auxiliary function $g$ influenced by the percentage of missing samples?  Both $\mathbf{L} _{v}$ and $\mathbf{P} _{v,s}$ incorporate the missing percentage, which raises the question of how this affects the function.

2. According to the dataset descriptions in Table 1, FASHMINST has more samples than YOUTUBEFACE. However, the results in Table 3 indicate that SIIHPC incurs a higher memory overhead on YOUTUBEFACE compared to FASHMINST. What factors contribute to this discrepancy?

3. As indicated in Table 3, GSRIMC is limited to operating on only two small-sized datasets and exhibits higher consumption of running time and memory overhead. In comparison with SIIHPC, what is the theoretical complexity of GSRIMC?

4. Beyond the comparison with graph or tensor-based methods, how does SIIHPC fare when compared to deep learning approaches?

**Details Of Ethics Concerns:**

N.A.

---

> ### Author Response · Authors · 2024-11-24
> **Reply for Reviewer 3rbh (1)**
>
> We would like to express our gratitude to Reviewer 3rbh  for their constructive comments and guidance regarding the revision of this paper. We have addressed all concerns in detail. We sincerely hope that all issues have been resolved.
>
> **Q1:**  The title of Section 5.5 is duplicated with that of Section G in the Appendix.
>
> **A1:**  We carefully proofread the manuscript and revise the title of Section G as 'Other Convergence Examples'.
>
>
> **Q2:**  Experiments exhibit that some comparison methods can not normally execute on VGGFACEHUND, YOUTUBEFACE and FASHMINST. There is a lack of some theoretical interpretations to support these observations.
>
> **A2:**    The methods LSIMVC, GSRIMC,  HCPIMSC, EEIMVC, LRGRIMVC, BGIMVSC, NGSPCGL, PIMVC and  HCLSCGL can not properly run   on VGGFACEHUND, YOUTUBEFACE or FASHMINST.  The reasons for this are as follows.
>
> To effectively eliminate the IMVC problem, these methods adopt various means. LSIMVC combines matrix factorization and sparse representation learning. GSRIMC integrates tensor nuclear norm,  cross-view relation learning  and bias sub-graph construction together.   HCPIMSC adopts the idea of tensor factorization, low-rank learning and hyper-Laplacian regularization. EEIMVC effectively merges multi-kernel learning and  consensus clustering learning.   Different from them, LRGRIMVC utilizes low-dimensional embedding and adaptive graph learning. BGIMVSC  unifies balanced spectral learning,  graph generation and  view adjustment.  NGSPCGL jointly conducts  neighbor group structure preserving and  consensus representation construction.  PIMVC performs graph projection, matrix factorization and  global geometric structure maintaining within one shared framework.  HCLSCGL embeds the Laplacian rank constraint  into the neighbor graph learning.  Although achieving effective clustering results from diverse aspects, these methods typically involve  constructing  $n \times n$ matrix,  which causes their complexity being at least $\mathcal{O}(n^2)$.  The intensive complexity limits their ability to tackle large-scale IMVC problems.
>
>
> **Q3:** The space complexity of Algorithm 2 is not described enough.
>
> **A3:**  Its space cost is mainly from storing $\mathbf{H} _{v,s}$, $\mathbf{G} _s$, $\mathbf{Q} _{v,s}$ and $\mathbf{A}$, $v=1, 2, \cdots, V; s=1, 2, \cdots, S$. In conjunction with the fact that  $\mathbf{H} _{v,s}$, $\mathbf{G} _s$, $\mathbf{Q} _{v,s}$ and $\mathbf{A}$ are in $\mathbb{R}^{d _v \times m _s}$, $\mathbb{R}^{m _s \times n}$,  $\mathbb{R}^{m _s \times n}$ and $\mathbb{R}^{V \times S}$ respectively,  we can get that it needs $\mathcal{O}(d _v m _s)$, $\mathcal{O}(m _s n)$, $\mathcal{O}(m _s n)$ and  $\mathcal{O}(VS)$ memory space respectively to store $\mathbf{H} _{v,s}$, $\mathbf{G} _s$, $\mathbf{Q} _{v,s}$ and $\mathbf{A}$.  Due to the numbers of views $V$ and prototype quantity candidates $S$ are constants, therefore, it totally needs  $\mathcal{O}(n \sum _{s=1}^{S} m _s + \sum _{v=1}^{V} \sum _{s=1}^{S} d _v m _s)$ memory cost to store all of them.  Additionally, the number of prototypes $m_s$ is largely smaller than the number of samples $n$, and the dimension of features $d_v$ is a constant and has nothing to do with $n$. 	Therefore, we have that the space complexity of Algorithm 2 is $\mathcal{O}(n)$.
>
>
> **Q4:** The definition/meaning of some notations is missing here, such as the $()_{+}$ in (11), N/A in Table 2 and Table 3.
>
> **A4:** Thanks! $(x) _{+}$ denotes that if $x>0$, it takes $x$ otherwise 0.  N/A denotes the running error caused by the complexity limit of algorithm itself.

---

> > ### Author Response · Authors · 2024-11-24
> > **Reply for Reviewer 3rbh (2)**
> >
> > **Q5:** Is the auxiliary function $g$ affected by the missing percentage? Based on $\mathbf{L} _{v}$ and $\mathbf{P} _{v,s}$, we can get that both of them contain the missing percentage.
> >
> > **A5:** The missing percentage affects the objective value of $g$ while not affecting  its monotonicity.
> >
> >
> > Combined with the definition of auxiliary function $g$, we can obtain that the missing percentage can affect the value of $g$ by changing $\mathbf{L} _{v}$  (equivalently, $\widehat{\mathbf{L}} _v$) and $\mathbf{P} _{v,s}$.
> >
> >
> > For any missing percentage, Lemma 1 always holds.  Although the missing percentage affects $\widehat{\mathbf{L}}_v$, it does not change Lemma 2.  Therefore, both Lemma 1 and Lemma 2 hold  under any missing percentage.   Accordingly, Theorem 1 holds under any missing percentage.   Therefore, we can get that the missing percentage does not affect the monotonicity of $g$.
> >
> >
> > **Q6:**  Given the introduction about the datasets in Table 1, one can get that FASHMINST has more samples than YOUTUBEFACE. However, the results in Table 3 show that SIIHPC requires more memory overhead on YOUTUBEFACE than FASHMINST. What factors caused this?
> >
> > **A6:**  This could be due to different numbers  of clusters on FASHMINST and YOUTUBEFACE.  In the paper, we introduce a group of hybrid prototype quantities $[1k,2k,3k,4k,5k]$ for each view to adaptively extract features where $k$ is the number of clusters. Accordingly, there will generate multi-scale  graphs with sizes $1k\times n$, $2k\times n$, $3k\times n$, $4k\times n$ and $5k\times n$ respectively on each view.  Besides, we construct consensus graphs, which are shared for all views, to provide unified structure, whose sizes are also $1k\times n$, $2k\times n$, $3k\times n$, $4k\times n$ and $5k\times n$ respectively.  So, on YOUTUBEFACE, it requires constructing 5 groups of $[20 \times 63896, 40 \times 63896, 60 \times 63896, 80 \times 63896, 100 \times 63896]$ graph matrices, while on FASHMINST,  it requires constructing 5 groups of $[10 \times 70000, 20 \times 70000, 30 \times 70000, 40 \times 70000, 50 \times 70000]$ graph matrices.  Given  $5 \times 20 \times 63896 > 5 \times 10 \times 70000$, therefore, we have that even though YOUTUBEFACE has fewer samples than FASHMINST, it requires more memory overhead.
> >
> >
> > **Q7:** Table 3 shows that GSRIMC only can work on two small-sized datasets and consumes more running time and memory overhead. In contrast with SIIHPC, what is its theoretical complexity.
> >
> > **A7:**  This is mainly caused by its intensive complexity.  Due to involving the construction of  unpolished graph tensor with size of $n \times n \times V$ , it needs at least $\mathcal{O} (n^2)$ space complexity.
> >
> > Besides, when solving sub-problems, it requires performing fast Fourier transformation and inverse fast Fourier transform on a  rotated tensor with size of $n \times V \times n$, which needs at least  $\mathcal{O} (n^2 \log(n) )$ computational  complexity.  Due to this, it generally is unable to work on large-scale datasets.

---

> > > ### Author Response · Authors · 2024-11-24
> > > **Reply for Reviewer 3rbh (3)**
> > >
> > > **Q8:**   In addition to the graph or tensor based methods, how does SIIHPC perform compared to deep methods?
> > >
> > > **A8:**  To further demonstrate the clustering ability of our SIIHPC, we  select the following three classical deep methods, APADC [1], CPM-Nets [2] and GP-MVC [3],  as the baselines.  The comparison results are presented in the following tables.
> > > 'BDG.','NUS.','VGY.' and 'VGD.' are  abbreviations  of BDGPFEA, NUSOBJECT, VGGFACEFIFTY and VGGFACEHUND respectively.
> > >
> > > |          |           |           |           |           | BDG.  |           |           |           |           |
> > > |:--------:|-----------|-----------|-----------|-----------|-----------|-----------|-----------|-----------|-----------|
> > > |          |           | 30\%      |           |           | 50\%      |           |           | 70\%      |           |
> > > |          | ACC       | NMI       | PUR       | ACC       | NMI       | PUR       | ACC       | NMI       | PUR       |
> > > | APADC    | 32.64     | 8.73      | 33.24     | 27.45     | 3.87      | 27.95     | 27.05     | 5.01      | 27.48     |
> > > | CPM-Nets | 35.56     | 14.62     | 36.92     | 27.98     | 7.71      | 29.34     | 29.78     | **12.64** | 33.02     |
> > > | GP-MVC   | 28.04     | 3.88      | 28.12     | 24.20     | 1.21      | 25.32     | 26.36     | 2.30      | 26.84     |
> > > | Ours     | **38.80** | **15.21** | **39.97** | **40.31** | **13.88** | **40.31** | **35.04** | 11.54     | **37.30** |
> > >
> > >
> > >
> > > |          |           |          |           |           | NUS. |           |           |          |           |
> > > |:--------:|:---------:|:--------:|:---------:|:---------:|:---------:|:---------:|:---------:|:--------:|:---------:|
> > > |          |           |   30\%   |           |           |    50\%   |           |           |   70\%   |           |
> > > |          |    ACC    |    NMI   |    PUR    |    ACC    |    NMI    |    PUR    |    ACC    |    NMI   |    PUR    |
> > > |   APADC  |   20.22   |   4.33   |   26.99   |   17.53   |    3.23   |   25.99   |   17.14   |   3.44   |   25.67   |
> > > | CPM-Nets |   22.39   |   6.88   |   29.01   |   21.18   |    5.97   |   27.82   |   20.24   |   4.60   |   26.15   |
> > > |  GP-MVC  |   16.80   |   3.20   |   26.16   |   16.91   |    5.40   |   27.36   |   17.90   |   7.51   |   30.84   |
> > > |   Ours   | **23.30** | **9.14** | **32.87** | **22.38** |  **9.21** | **33.92** | **21.46** | **8.36** | **31.49** |
> > >
> > >
> > >
> > >
> > > |          |           |           |           |           | VGY. |           |           |           |           |
> > > |:--------:|:---------:|:---------:|:---------:|:---------:|:------------:|:---------:|:---------:|:---------:|:---------:|
> > > |          |           |    30\%   |           |           |     50\%     |           |           |    70\%   |           |
> > > |          |    ACC    |    NMI    |    PUR    |    ACC    |      NMI     |    PUR    |    ACC    |    NMI    |    PUR    |
> > > |   APADC  |    5.27   |    4.09   |    5.89   |    5.11   |     4.06     |    5.71   |    4.96   |    3.86   |    5.60   |
> > > | CPM-Nets |    9.00   |   12.44   |   10.41   |    8.78   |     12.08    |   10.20   |    8.01   |   10.70   |    9.34   |
> > > |  GP-MVC  |   10.42   |   13.46   |   11.51   |    9.46   |     11.87    |   10.45   |    8.86   |   10.97   |    9.74   |
> > > |   Ours   | **12.52** | **14.91** | **13.44** | **12.31** |   **14.48**  | **13.21** | **11.18** | **13.35** | **11.96** |
> > >
> > >
> > >
> > >
> > > |          |          |           |          |          | VGD.  |          |          |           |          |
> > > |:--------:|:--------:|:---------:|:--------:|:--------:|:------------:|:--------:|:--------:|:---------:|:--------:|
> > > |          |          |    30\%   |          |          |     50\%     |          |          |    70\%   |          |
> > > |          |    ACC   |    NMI    |    PUR   |    ACC   |      NMI     |    PUR   |    ACC   |    NMI    |    PUR   |
> > > |   APADC  |   4.15   |    9.60   |   4.31   |   4.14   |     9.00     |   4.37   |   3.90   |    6.91   |   4.12   |
> > > | CPM-Nets |   5.42   |   12.56   |   6.35   |   5.47   |     12.57    |   6.48   |   5.37   |   11.78   |   6.22   |
> > > |  GP-MVC  |   6.26   |   13.45   |   7.39   |   5.53   |     11.51    |   6.38   |   4.59   |   10.62   |   5.51   |
> > > |   Ours   | **8.26** | **14.94** | **9.13** | **7.55** |   **13.85**  | **8.39** | **6.82** | **12.69** | **7.55** |
> > >
> > >
> > >
> > >
> > >
> > > As seen, our SIIHPC still can receive preferable clustering results against them in most cases.
> > >
> > >
> > > 	[1]  J. Xu, et al., Adaptive Feature Projection With Distribution Alignment for Deep Incomplete Multi-View Clustering, IEEE TIP, 2023.
> > >
> > > 	[2]  C. Zhang et al., Deep Partial Multi-View Learning, IEEE TPAMI, 2022.
> > >
> > > 	[3]  Q. Wang et al., Generative Partial Multi-View Clustering with Adaptive Fusion and Cycle Consistency, IEEE TIP, 2021.

---

> > > > ### Author Response · Authors · 2024-11-28
> > > > **Reply for Reviewer 3rbh**
> > > >
> > > > Dear reviewer 3rbh,
> > > >
> > > > We deeply appreciate the time you took to provide your thoughtful and profound  feedback, and sincerely hope your concerns have been addressed in our response. If you have any additional suggestions, please feel free to contact us. We would be very happy to engage in a discussion with you. We understand that your time is valuable and appreciate your willingness to participate in this process.
> > > >
> > > >
> > > >
> > > > Best regards,
> > > >
> > > > Authors of paper 778

---

> > > > ### Comment · Reviewer_3rbh · 2024-11-28
> > > >
> > > > Thank you for the authors' rebuttal. My concerns have been addressed.

---

> > > > > ### Author Response · Authors · 2024-11-29
> > > > > **Reply for Reviewer 3rbh**
> > > > >
> > > > > Thank you for your support!

---

### Official Review · Reviewer_fE1F · 2024-11-04

**Soundness:** 3
**Presentation:** 3
**Contribution:** 3
**Rating:** 6
**Confidence:** 4

**Summary:**

This paper introduces SIIHPC, an incomplete multi-view clustering method that bypasses traditional missing view recovery by directly imputing similarities. SIIHPC replaces single prototype construction with hybrid-group prototypes, allowing each view to capture unique features within a unified framework. This framework is optimized through a well-designed iterative algorithm with convergence guarantees, ensuring robust and effective clustering.

**Strengths:**

The paper presents a novel approach SIIHPC to incomplete multi-view clustering through similarity-level imputation, which diverges from conventional methods that focus on reconstructing missing samples or features. This approach serves as an efficient alternative to traditional recovery-based techniques. By addressing incomplete multi-view clustering without explicit missing view imputation, SIIHPC reduces computational overhead and allows clustering methods to function effectively without relying on fully complete data structures.

A key innovation of the paper is the introduction of hybrid-group prototype construction, which replaces single prototypes with a more flexible model that enables each view to capture unique features. This enhances the model’s adaptability to heterogeneous data views, improving clustering performance across diverse datasets and scenarios.

Extensive experimental evaluations on multiple datasets with varying missing ratios highlight the robustness and effectiveness of SIIHPC. The inclusion of ablation studies further enriches the assessment, providing valuable insights into the contribution of each component to the model's overall performance and validating the framework’s design choices.

**Weaknesses:**

The terminology introduced in the paper is unclear; for instance, terms like T-PBL, SLI, IVHGP, and MSVSG are not properly explained in the captions of Figure 1, making it difficult for readers to follow the framework without additional context.

The overall process of the proposed clustering algorithm lacks clarity. Specifically, the final steps for obtaining clustering results and the initialization settings for relevant parameters should be clearly defined to improve the reader's understanding of the algorithm’s workflow.

While the paper provides strong experimental results, the interpretability of the clustering outputs and the practical implications of hybrid-group prototypes could be explored further. It would be helpful for the authors to discuss how the clustering results or prototype structures might be interpreted in real-world applications, providing insights into the method's broader utility.

**Questions:**

Can the authors clarify the terms used in Figure 1, such as T-PBL, SLI, IVHGP, and MSVSG, and ensure that these terms are defined either in the figure caption or in the text to improve readability?

Could the authors provide a more detailed explanation of the overall process of the clustering algorithm, particularly the final steps for obtaining clustering results and the initial parameter settings?

Given that the method relies on similarity-level imputation rather than missing view recovery, how sensitive is the model's performance to the quality of the imputed similarities? Could the authors discuss any potential limitations in scenarios with high levels of data incompleteness?

To better assess the stability and reliability of SIIHPC’s performance, could the authors provide standard deviation metrics for their experimental results? Including this information would help illustrate the consistency of the method across different runs and give a clearer view of its robustness under varying data conditions.

---

> ### Author Response · Authors · 2024-11-24
> **Reply for Reviewer fE1F  (1)**
>
> We greatly appreciate the profound comments provided by Reviewer fE1F  for the revision of this manuscript. We have addressed all concerns in detail and hope that all issues have been successfully resolved.
>
> **Q1:** Can the authors clarify the terms used in Figure 1, such as T-PBL, SLI, IVHGP, and MSVSG, and ensure that these terms are defined either in the figure caption or in the text to improve readability?
>
> **A1:** We have updated Figure 1, please check it.  Specially,
>
> * T-PBL represents transforming partial bipartition learning into  original sample form to split out of observed similarity.
>
> * IVHGP represents introducing a group of hybrid prototype quantities for each individual view to flexibly extract the data features belonging to each view itself, i.e.,  intra-view hybrid-group prototypes.
>
> * OS represents the observed similarity on each individual view.
>
> * MSVSG represents the generated  graphs  with various scales on each view, i.e., multi-scale view-specific graphs.
>
> * SLI represents learning to recover the incomplete parts by utilizing the connection built between the similarity exclusive on respective view and the consensus graph shared for all views,	i.e., similarity-level imputation.
>
> * CG represents the learned consensus graph that is shared for all views.
>
> * SG represents conducting spectral grouping on the graph generated by stacking multi-scale consensus graphs.
>
> These detailed descriptions are also added in Section C of the appendix.
>
>
> **Q2:**  Could the authors provide a more detailed explanation of the overall process of the clustering algorithm, particularly the final steps for obtaining clustering results and the initial parameter settings?
>
> **A2:**  We here provide a more detailed explanation about the overall process of the clustering algorithm.
>
> To be specific,  we first initialize the prototype matrix $\mathbf{H} _{v,s}$ with the $s$-th quantity on view $v$  using the orthogonal matrix, the imputation matrix $\mathbf{Q} _{v,s}$  with the $s$-th scale on view $v$  using the random matrix with elements ranging from -1 to 1,  the consensus graph $\mathbf{G} _s$ with the $s$-th scale using the random matrix with elements ranging from -1 to 1, the prototype quantity coefficient matrix $\mathbf{A}$ using ${1}/{S}$ where $S$ denotes the number of hybrid prototype quantity candidates.
>
> Then, we perform Algorithm 2 until satisfying $(f _{obj} (t)-f _{obj} (t + 1))/f _{obj} (t) <= 1e-4$ where $f _{obj} (t)$ denotes the objective value at the $t$-th iteration.
>
> Subsequently, we stack all generated unified representation matrices $\\{ \mathbf{G} _s\\} _{s=1}^{S}$ by rows, and  conduct spectral grouping on it to obtain the final clustering results.
>
>
>
> **Q3:** 	Given that the method relies on similarity-level imputation rather than missing view recovery, how sensitive is the model's performance to the quality of the imputed similarities? Could the authors discuss any potential limitations in scenarios with high levels of data incompleteness?
>
> **A3(--1):**  	In our model, the incomplete similarity is adaptively imputed by comparing  the learned consensus graphs and multi-scale view-specific graphs. It alleviates the  cluster imbalance caused by the he unpairing of observed samples, and also makes information from different views able to communicate with each other so as to more accurately construct similarity.
>
> To investigate its effect, we organize some comparison experiments and the results are summarized in the following table where 'NSLF' and 'WSLF'  represent the results with/without our similarity-level imputation, respectively.  'BDG.','NUS.','VGY.','VGD.','YOU.' and 'FAS.' are  abbreviations  of BDGPFEA, NUSOBJECT, VGGFACEFIFTY, VGGFACEHUND, YOUTUBEFACE and FASHMINST, respectively.

---

> > ### Author Response · Authors · 2024-11-24
> > **Reply for Reviewer fE1F (2)**
> >
> > **A3(--2):**
> >
> > |  |  |  |  |  | **BDG.** |  |  |  |  |
> > |:---:|:---:|:---:|:---:|:---:|:---:|:---:|:---:|:---:|:---:|
> > |  |  | 30\% |  |  | 50\% |  |  | 70\% |  |
> > |  | ACC | NMI | PUR | ACC | NMI | PUR | ACC | NMI | PUR |
> > | NSLF | 28.15 | 3.83 | 30.38 | 29.74 | 3.97 | 30.98 | 26.39 | 1.68 | 26.80 |
> > | WSLF | **38.80** | **15.21** | **39.97** | **40.31** | **13.88** | **40.31** | **35.04** | **11.54** | **37.30** |
> > |  |  |  |  |  | **NUS.** |  |  |  |  |
> > | NSLF | 22.60 | 7.29 | 31.85 | 21.09 | 6.20 | 31.10 | 18.05 | 3.25 | 27.86 |
> > | WSLF | **23.30** | **9.14** | **32.87** | **22.38** | **9.21** | **33.92** | **21.46** | **8.36** | **31.49** |
> > |  |  |  |  |  | **VGY.** |  |  |  |  |
> > | NSLF | 6.71 | 7.22 | 7.51 | 5.33 | 4.45 | 6.00 | 5.01 | 3.76 | 5.64 |
> > | WSLF | **12.52** | **14.91** | **13.44** | **12.31** | **14.48** | **13.21** | **11.18** | **13.35** | **11.96** |
> > |  |  |  |  |  | **VGD.** |  |  |  |  |
> > | NSLF | 4.83 | 9.93 | 5.53 | 3.73 | 6.69 | 4.26 | 3.12 | 4.98 | 3.56 |
> > | WSLF | **8.26** | **14.94** | **9.13** | **7.55** | **13.85** | **8.39** | **6.82** | **12.69** | **7.55** |
> > |  |  |  |  |  | **YOU.** |  |  |  |  |
> > | NSLF | 46.19 | 40.95 | 51.69 | 26.07 | 16.16 | 28.60 | 15.82 | 15.40 | 17.46 |
> > | WSLF | **76.29** | **82.27** | **80.81** | **72.60** | **79.60** | **77.65** | **71.05** | **79.19** | **76.75** |
> > |  |  |  |  |  | **FAS.** |  |  |  |  |
> > | NSLF | 46.99 | 33.74 | 49.41 | 37.85 | 24.03 | 40.85 | 25.15 | 9.26 | 27.31 |
> > | WSLF | **61.24** | **59.52** | **62.69** | **62.51** | **60.22** | **64.64** | **60.59** | **58.77** | **63.18** |
> >
> >
> >
> > It can be seen that our similarity-level imputation significantly improves the clustering results.  Additionally, after imputation, the performance does not fluctuate drastically with the missing ratio.
> >
> >
> > Further, we also explore its effect when the missing ratio is up to 80\%. Please see the following table.
> >
> > |  |  |  |  |  | **BDG.** |  |  |  |  |
> > |:---:|:---:|:---:|:---:|:---:|:---:|:---:|:---:|:---:|:---:|
> > |  |  | 40\% |  |  | 60\% |  |  | 80\% |  |
> > |  | ACC | NMI | PUR | ACC | NMI | PUR | ACC | NMI | PUR |
> > | NSLF | 32.59 | 5.37 | 33.83 | 27.83 | 3.03 | 28.91 | 24.40 | 1.04 | 24.69 |
> > | WSLF | **41.03** | **14.27** | **42.43** | **36.13** | **12.09** | **38.13** | **36.79** | **10.71** | **37.91** |
> > |  |  |  |  |  | **NUS.** |  |  |  |  |
> > | NSLF | 21.63 | 6.44 | 31.23 | 19.72 | 4.55 | 28.73 | 16.97 | 2.78 | 27.17 |
> > | WSLF | **22.32** | **8.50** | **31.97** | **21.99** | **8.16** | **32.15** | **21.38** | **7.19** | **31.75** |
> > |  |  |  |  |  | **VGY.** |  |  |  |  |
> > | NSLF | 7.18 | 8.52 | 7.96 | 5.04 | 3.88 | 5.70 | 4.22 | 2.56 | 4.81 |
> > | WSLF | **12.68** | **14.85** | **13.45** | **11.65** | **13.83** | **12.53** | **11.20** | **13.17** | **12.03** |
> > |  |  |  |  |  | **VGD.** |  |  |  |  |
> > | NSLF | 4.43 | 9.19 | 5.07 | 3.45 | 5.92 | 3.90 | 2.79 | 4.16 | 3.19 |
> > | WSLF | **7.77** | **14.26** | **8.60** | **7.14** | **13.18** | **7.88** | **6.60** | **12.12** | **7.37** |
> > |  |  |  |  |  | **YOU.** |  |  |  |  |
> > | NSLF | 32.95 | 24.62 | 37.19 | 22.02 | 10.41 | 25.10 | 13.42 | 3.58 | 15.01 |
> > | WSLF | **75.34** | **79.89** | **79.97** | **74.98** | **80.49** | **79.68** | **70.40** | **77.78** | **75.72** |
> > |  |  |  |  |  | **FAS.** |  |  |  |  |
> > | NSLF | 40.46 | 29.48 | 43.62 | 29.63 | 16.36 | 32.63 | 21.38 | 7.41 | 22.84 |
> > | WSLF | **61.92** | **60.86** | **65.38** | **57.95** | **59.49** | **60.94** | **66.15** | **60.96** | **66.15** |
> >
> >
> >
> > As seen, our similarity-level imputation is still functional, which is mainly owing to the fact that our clustering model learns to adaptively recover the similarity.
> >
> >
> >
> > About the potential limitations in scenarios with high levels of data incompleteness,   in this work, we adopt cosine tool to do similarity measure, other more sensible measure means could  be  investigated to further enhance the clustering performance of SIIHPC.
> > Moreover, we aggregate the information of generated graph similarities with diverse scales by stacking them and performing SVD to construct splicing eigenvectors. This could weaken the contributions of certain graphs or certain eigen-components. Especially, when encountering high levels of data incompleteness,  the importance differences between graphs or eigen-components will be further enlarged.

---

> > > ### Author Response · Authors · 2024-11-24
> > > **Reply for Reviewer fE1F (3)**
> > >
> > > **Q4:** To better assess the stability and reliability of SIIHPC’s performance, could the authors provide standard deviation metrics for their experimental results?
> > >
> > > **A4:**   Very valuable suggestion! We count the standard deviation (\%) in the following table.
> > >
> > >
> > > |  |  |  |  | **BDG.** |  |  |  |  |
> > > |:---:|:---:|:---:|:---:|:---:|:---:|:---:|:---:|:---:|
> > > |  | 30\% |  |  | 50\% |  |  | 70\% |  |
> > > | ACC | NMI | PUR | ACC | NMI | PUR | ACC | NMI | PUR |
> > > | 0.25 | 0.16 | 0.37 | 0.24 | 0.02 | 0.24 | 0.32 | 0.01 | 0.06 |
> > > |  |  |  |  | **NUS.** |  |  |  |  |
> > > | 0.30 | 0.13 | 0.17 | 0.09 | 0.14 | 0.20 | 0.13 | 0.06 | 0.16 |
> > > |  |  |  |  | **VGY.** |  |  |  |  |
> > > | 0.30 | 0.35 | 0.33 | 0.32 | 0.31 | 0.27 | 0.36 | 0.31 | 0.29 |
> > > |  |  |  |  | **VGD.** |  |  |  |  |
> > > | 0.25 | 0.25 | 0.25 | 0.24 | 0.16 | 0.25 | 0.19 | 0.22 | 0.18 |
> > > |  |  |  |  | **YOU.** |  |  |  |  |
> > > | 2.46 | 1.07 | 1.58 | 2.09 | 1.30 | 1.88 | 2.55 | 0.82 | 1.67 |
> > > |  |  |  |  | **FAS.** |  |  |  |  |
> > > | 0.00 | 0.00 | 0.00 | 0.00 | 0.01 | 0.01 | 0.00 | 0.28 | 0.00 |
> > >
> > >
> > > One can observe that even under diverse missing rations, our standard deviation
> > > is  small, which illustrates that our SIIHPC is stable under varying data conditions.
> > >
> > > To further demonstrate this, we also count the standard deviation under missing ratio being 40\%, 60\% and 80\%, as shown in the following table.
> > >
> > > |      |      |      |      | **BDG.** |      |      |      |      |
> > > |:----:|:----:|:----:|:----:|:--------:|:----:|:----:|:----:|:----:|
> > > |      | 40\% |      |      |   60\%   |      |      | 80\% |      |
> > > |  ACC |  NMI |  PUR |  ACC |    NMI   |  PUR |  ACC |  NMI |  PUR |
> > > | 0.25 | 0.16 | 0.37 | 0.24 |   0.02   | 0.24 | 0.32 | 0.01 | 0.06 |
> > > |      |      |      |      | **NUS.** |      |      |      |      |
> > > | 0.30 | 0.13 | 0.17 | 0.09 |   0.14   | 0.20 | 0.13 | 0.06 | 0.16 |
> > > |      |      |      |      | **VGY.** |      |      |      |      |
> > > | 0.30 | 0.35 | 0.33 | 0.32 |   0.31   | 0.27 | 0.36 | 0.31 | 0.29 |
> > > |      |      |      |      | **VGD.** |      |      |      |      |
> > > | 0.25 | 0.25 | 0.25 | 0.24 |   0.16   | 0.25 | 0.19 | 0.22 | 0.18 |
> > > |      |      |      |      | **YOU.** |      |      |      |      |
> > > | 2.46 | 1.07 | 1.58 | 2.09 |   1.30   | 1.88 | 2.55 | 0.82 | 1.67 |
> > > |      |      |      |      | **FAS.** |      |      |      |      |
> > > | 0.00 | 0.00 | 0.00 | 0.00 |   0.01   | 0.01 | 0.00 | 0.28 | 0.00 |
> > >
> > >
> > > Evidently, our standard deviation is still small.
> > >
> > >
> > >
> > > These experimental results are also added in Section M of the appendix.
> > > Please check it.

---

> > ### Comment · Reviewer_fE1F · 2024-11-26
> >
> > After reviewing the rebuttal, I appreciate the authors' efforts and will raise my score.

---

> > > ### Author Response · Authors · 2024-11-27
> > > **Reply for Reviewer fE1F**
> > >
> > > Thanks!!!

---

### Author Response · Authors · 2024-11-25
**Reply for SAC, AC, Reviewers**

Dear SAC, AC, Reviewers,


We would like to extend our heartfelt thanks to SAC and AC as well as Reviewers for the dedicated efforts and invaluable feedback.   Your insightful comments and constructive criticisms have not only helped us improve the quality of our work but also offered us new perspectives and ideas that we had not considered before.  We are truly grateful for your meticulous review and the opportunity to revise our paper based on your suggestions.

This manuscript mainly focuses on two issues in incomplete multi-view clustering (IMVC): the ignoring of missing samples and the single quantity of prototypes.



Most of existing IMVC  methods typically choose to ignore the missing samples and only utilize the observed samples to  achieve the construction of bipartite similarity.   This will miss out latent useful information from the missing samples, resulting in the generated similarity not that accurate. Also, due to the randomness of sample missing, the remaining observed samples are usually unpaired, which could lead to unbalanced cluster distribution and deteriorate the graph structure.  To this end,  we firstly transform partial bipartition learning into original sample form by virtue of reconstruction concept to split out of observed similarity, and then loosens traditional non-negative constraints via regularizing samples to more freely characterize the similarity. Afterwards, we  learn to recover the incomplete parts by utilizing the connection built between the similarity exclusive on respective view and the consensus graph shared for all views.



Subsequently, different from current techniques that employ a single quantity of prototypes to extract the information of all views,  we introduce the intra-view hybrid-group prototype quantity learning paradigm  to flexibly extract the data features belonging to each view itself.  It can automatically select the appropriate quantity of prototypes according to view characteristics.  Consequently, the resulting graphs are with various scales and describe the overall similarity more comprehensively.


Further, to optimize the formulated objective loss,  we design an ingenious auxiliary function with  theoretically and experimentally proven monotonic-increasing properties.


It is worth mentioning that these components are all in one unified framework,  which enables them to mutually support and encourage during the learning process.  Meanwhile, the overall linear complexity guarantees that our model can be extended to large-scale tasks.

We also count the clustering variance under diverse missing ratios, and the experimental results suggest that our proposed SIIHPC is stable and reliable  under varying data conditions.

Moreover, the algorithm convergence, the hyper-parameter sensitivity and the hybrid prototype quantity influence are also well explored in the manuscript.


Experiments on several public datasets demonstrate that even encountering diverse missing proportions, our SIIHPC is still able to provide a preferable clustering performance compared to multiple remarkable IMVC competitors.




We hope the above statements have addressed your concerns, and highly appreciate your thoughtful and profound reviews.  If any questions, please feel free to contact us! We are fairly happy to further discuss them.



Best regards,

Authors of paper 778

---

### Meta-Review · Area_Chair_iK4Z · 2024-12-18

**Metareview:**

The paper aims to tackle the missing sample problem in multi-view clustering task. The problem itself is important and valuable in the real-world scenarios. The proposed approach is technically sound. The extensive empirical studies, both in the manuscript and appendix, are thorough and convincing.

The paper received three clear acceptances and one borderline acceptance from four expert reviewers. I agree with the reviewers’ assessments and recommend acceptance.

**Additional Comments On Reviewer Discussion:**

The main concerns raised by the reviewers involve some unclear statements and details of the approach, as well as the sensitivity of performance. During the rebuttal period, the authors effectively addressed these concerns by clarifying ambiguities and providing additional results, which strengthened the paper’s contributions.

---

### Decision · Program_Chairs · 2025-01-22

Accept (Spotlight)